# Capturing ultrafast molecular motions and lattice dynamics in spin crossover film using femtosecond diffraction methods

Doriana Vinci[1], Karl Ridier [2], Fengfeng Qi[3,4], Fernando Ardana-Lamas[1], Peter Zalden[1], Lai Chung Liu [5], Tobias Eklund[1,6], Mads Sielemann Jakobsen[7,8], Robin Schubert [1], Dmitry Khakhulin[1], Carsten Deiter [1], Nicolas Bottin[1], Hazem Yousef[1], David von Stetten [9], Piotr Łaski[10], Radosław Kamiński [10], Katarzyna N. Jarzembska [10], Rachel F. Wallick[11], Till Stensitzki[12], Renske M. van der Veen [11,13,14], Henrike M. Müller-Werkmeister [12], Gábor Molnár[2], Dao Xiang [3,4,15] ✉, Christopher Milne [1] ✉, Maciej Lorenc [16,17] ✉ & Yifeng Jiang [1] ✉

A comprehensive insight into ultrafast dynamics of photo-switchable materials is desired for efficient control of material properties through light excitation. Here, we study a polycrystalline spin crossover thin film as a prototypical example and reveal the sequential photo-switching dynamics, from local molecular rearrangement to global lattice deformation. On the earliest femtosecond timescale, the local molecular structural rearrangement occurs within a constant unit-cell volume through a two-step process, involving initial Fe–ligand bond elongation followed by ligand rotation. The highly-oriented structure of the nanocrystalline films and the experimental geometry enables resolving the full anisotropic lattice structural dynamics in and out of the sample plane separately. While both molecular switching and lattice heating influence lattice volume, they exert varying degrees of impact at disparate time scales following photoexcitation. This study highlights the opportunities provided by Mega-electron-volt electron and X-ray free electron laser to advance the understanding of ultrafast dynamics of photo-switchable materials.

In the realms of crystal engineering, molecular electronics and supramolecular chemistry, photoinduced phase transitions have attracted great attention as they provide a way to control various photoswitchable functions (*e.g.* optical properties, conductivity, magnetism, and dielectric response)[1–5]. Exploring time-resolved molecular structural changes and deformations of the crystallographic lattice (unit-cell symmetry, parameters, volume) during photoinduced phase transitions is critical to understand the pathways through which photoinduced energy is transferred between electronic and vibrational degrees of freedom[6]. This knowledge provides understanding of the non-equilibrium dynamics on the atomic length scale and the interplay between the intricate processes that govern the behavior of matter at its most fundamental level[5,7–9].

From the perspective of ultrafast structural dynamics, photoswitchable spin crossover (SCO) materials are particularly attractive platforms due to their functional design and tuning versatility on both the molecular and the crystal structure levels[10,11]. The switching dynamics in SCO nanomaterials following light excitation involve two

A full list of affiliations appears at the end of the paper. ✉e-mail: dxiang@sjtu.edu.cn; christopher.milne@xfel.eu; maciej.lorenc@univ-rennes1.fr; yifeng.jiang@xfel.eu

sequential processes: an ultrafast photoinduced SCO occurring on the sub-picosecond (sub-ps) timescale, followed by a second thermo-elastic SCO step on a longer timescale, in the nanosecond (ns) range[6,12–16]. The ultrafast photoinduced SCO dynamics has been extensively studied using multiple methods[6,13,14,17–31] and some consensus about the general photoconversion pathways has been achieved, though these pathways are strongly affected by the local coordination environment of the metal centers. Upon photon absorption, the SCO molecule is excited from the low-spin (LS) state into a charge-transfer manifold and then relaxes to a high-spin (HS) state, during which a rapid intramolecular vibrational energy redistribution (IVR) occurs on the hundreds of femtoseconds (fs) timescale[16–18,22–24,31,32]. The SCO photoswitching process has been described by qualitatively splitting the reaction coordinate into the symmetric breathing mode of the metal-ligand bonds and local ligand bending mode[19,20,25,30]. The associated metal-ligand bond length elongation and structural distortions of the ligand provides the possibility of feedback occurring between the expanding lattice volume and molecular spin state[6,13]. Indeed, through intermolecular interactions in the crystal phase, local excitations can propagate and lead to a global lattice volume change via the propagation of strain waves[14,33]. In addition, as a large part of the photon energy is transferred to the lattice, a sizeable rise of the lattice temperature occurs on a timescale of tens of ps[14,34]. The lattice expansion allows a further structural reorganization, eventually yielding a fully relaxed HS structure similar to the thermally-induced HS state[35,36]. As a consequence, a delayed thermo-elastic driven SCO proceeds on longer timescales (tens of ns), governed by the intramolecular energy barrier between the LS and HS states, since the excitation fraction of molecules equilibrates within the new lattice conformation[14–16]. However, the exact time evolution and interaction between molecular structural rearrangement, photo-thermal heating, change in the unit-cell volume, and subsequential spin-state switching in (nano-)crystals on ultrafast timescales remains a subject of ongoing debate. More specifically, thermal and elastic (volume change, internal pressure) effects are known to play a crucial role in the lattice dynamics, thus it appears vital to disentangle these different contributions in the out-of-equilibrium switching dynamics of SCO nanomaterials. Understanding these thermo-elastic effects is, however, not sufficient because the SCO process is also governed by local and bulk (whole crystal) energy barriers, which can further perturb the collective electronic and structural rearrangements of the molecules. An important issue is thus to explore how such couplings might facilitate or impede cooperative, fast, and efficient bulk phase transformations[13,37], in order to achieve (as for all photoinduced phase transition materials) various photo-switchable functions driven by short light excitations[10,11].

Recent developments in the nanoscale synthesis of SCO materials have enabled the fabrication of a variety of molecular nanoparticles and thin films exhibiting a spin transition at technologically relevant temperatures[38,39]. A recently synthesized SCO system is a thin film of the iron(II) molecular complex [Fe(HB(tz)$_3$)$_2$] (tz = 1,2,4-triazol-1-yl) (**1**) which demonstrates isostructural (orthorhombic *Pbca* space group) SCO above room temperature ($T_C$ = 336 K) with exceptionally high resilience upon repeated (photo-)switching[16,40–43]. The molecular structure of **1** in the LS state is presented in Fig. 1a. One specific structural characteristic of **1** is its opposing evolution of the *a* unit-cell parameter during the molecular switching and thermal expansion. In Fig. 1b, the *a* unit-cell parameter decreases by 2.3 % upon switching from the LS state (low-temperature (LT) phase, *T* = 300 K) to the HS state (high-temperature (HT) phase, *T* = 373 K), while the *b* and *c* unit-cell parameters increase by 1.0 and 5.6 %, respectively, as reported in previous studies[15,44]. Far from the spin-transition temperature ($T_C$ = 336 K), all three unit-cell parameters increase with temperature due to ordinary thermal expansion with typical linear dilatation coefficients in the range 3–8 × 10$^{-4}$ Å K$^{-1}$ [15,44]. Furthermore, the SCO

complex **1** can be deposited by vacuum thermal evaporation on different substrates to obtain smooth, dense, highly-oriented thin films with the orthorhombic *Z* crystallographic direction normal to the surface[38]. The crystallite size in the sample plane is on the order of tens of μm, similar to the size of the electron and X-ray beams in our experiment, allowing us to obtain single-crystal diffraction patterns and study Bragg peaks with similar inter-planar spacing (*d*) separately (Fig. 1c). The photo-response of films of **1** has been investigated by transient optical absorption spectroscopy[16] and grazing-incidence X-ray diffraction (XRD) at the European Synchrotron Radiation Facility (ESRF)[15], but these studies lacked either the temporal resolution or structural sensitivity to reveal the mechanistic details of the molecular and crystal lattice rearrangements on the ultrafast timescales.

In this work, we show experimental data obtained by two complementary time-resolved structural methods: ultrafast electron diffraction (UED) performed at the Shanghai Jiao Tong University[45] (Shanghai, China) and ultrafast XRD[46,47] conducted at the FXE instrument of the European XFEL (EuXFEL) (Schenefeld, Germany), as

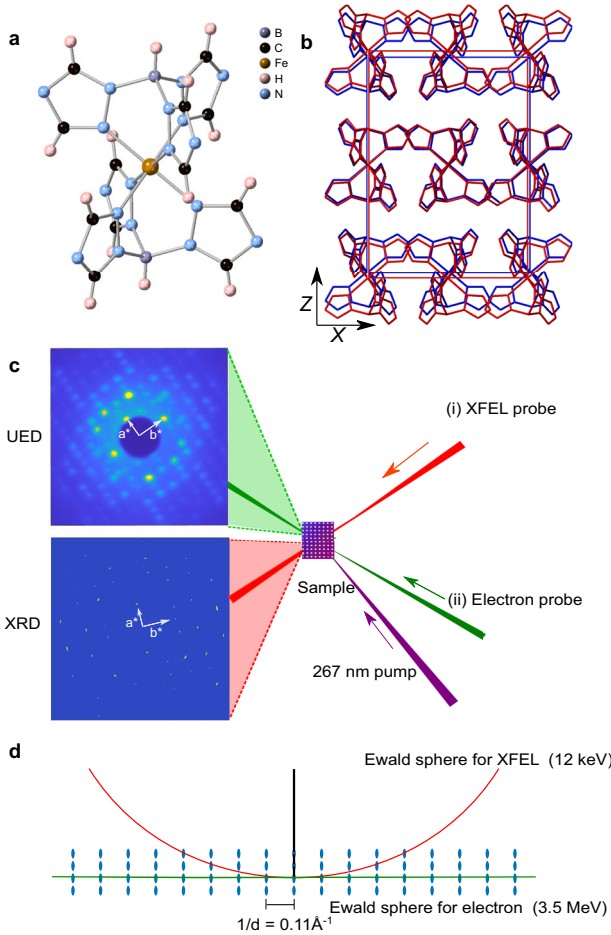

**Fig. 1 | Schematic structure, experimental setup, and Ewald sphere. a** Molecular structure of the complex **1** in the LS state. Black balls are carbon atoms; blue balls are nitrogen atoms; brown balls are iron atoms; purple balls are boron atoms; pink balls are hydrogen atoms; **b** Unit-cell and molecular structure of **1** in the LT (*T* = 300 K) (blue bonds and unit-cell edges) and HT (*T* = 373 K) (red bonds and unit-cell edges) phases. Hydrogen atoms are omitted for clarity. **c** Schematic of the experimental setup and representative diffraction images. After photoexcitation with 267-nm pump pulses, the structural response of **1** was measured separately by (i) ultrafast XRD at an XFEL and (ii) MeV-UED. **d** Construction of the Ewald sphere for the XFEL (red) and MeV UED (green) experiments. The specific energies of the keV XFEL and MeV electron sources imply vastly different radii of the Ewald spheres for the XRD and UED experiments, respectively.

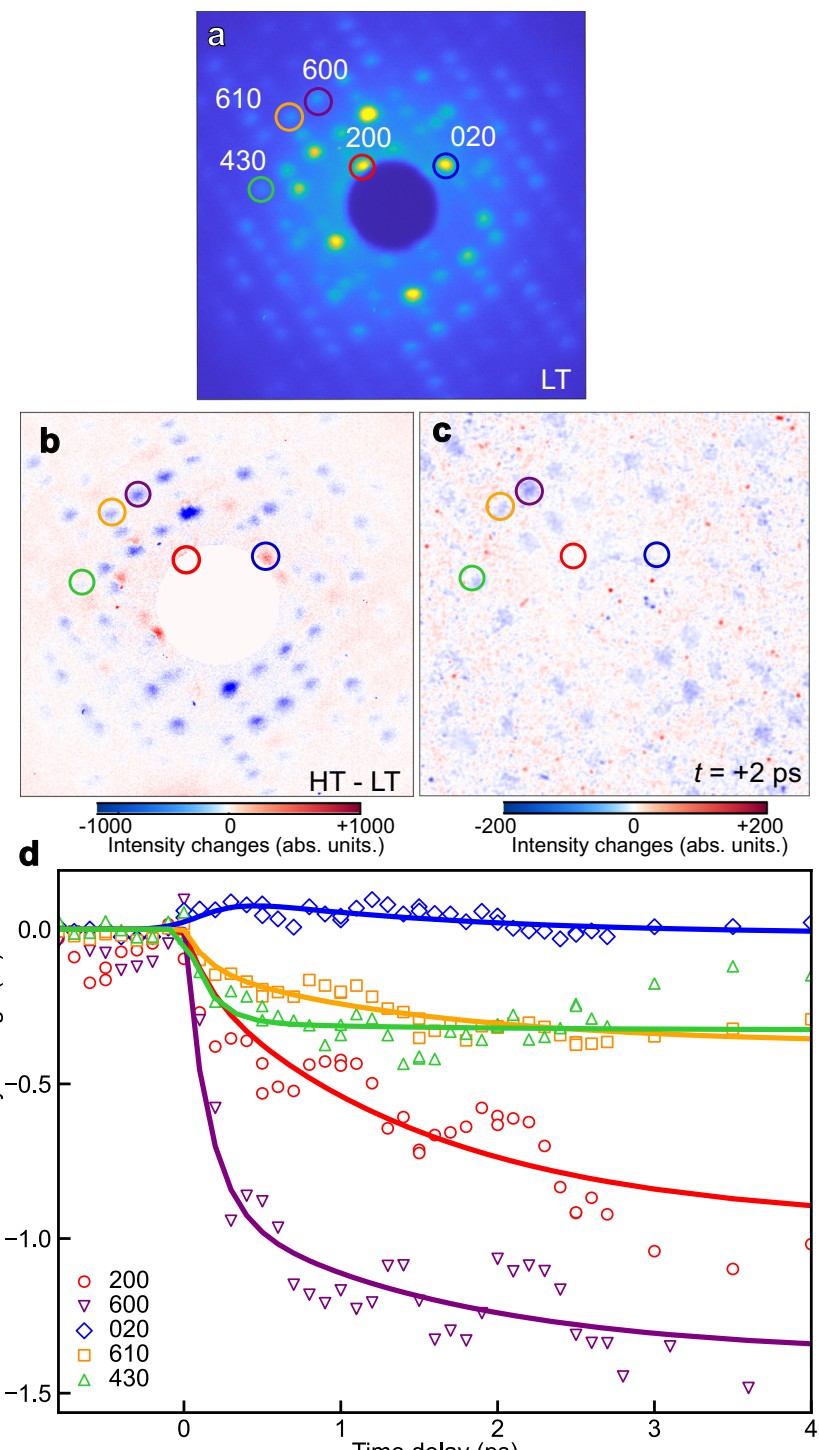

**Fig. 2 | Ultrafast electron diffraction study. a** Static electron diffraction pattern at 300 K (LT) phase. **b** Difference between the diffraction patterns in LT and HT (373 K) phases. The differences near the beam stop are blocked due to bulk motion of the sample. **c** Photoinduced changes in the diffraction pattern measured at + 2 ps after photoexcitation at 267 nm at 300 K. **a–c** Color circles show the positions of the selected Bragg peaks. **d** Kinetic traces of Bragg peak intensity for selected reflections from − 1 to + 4 ps. The solid lines show the results of the global fit to a biexponential decay functions (Supplementary Eq. 2).

## Results and Discussion

### Ultrafast molecular switching

UED enables direct determination of atomic positions in crystals by scattering electrons off the Coulomb potentials of nuclei with fs temporal resolution. In Fig. 2a, a static electron diffraction pattern of **1** in the LS state at 300 K is presented with a $q$ range of 0.92–4.7 Å$^{-1}$ (note $q = 4\pi \sin\theta/\lambda$, with $\lambda$ the wavelength of electrons and $2\theta$ the diffraction angle). Owing to the large momentum of MeV electrons, the Ewald

schematically illustrated in Fig. 1c, d. The combination of these two diffraction techniques allows us to monitor the molecular motions and the unit-cell volume changes in SCO thin films with sub-100 fs temporal resolution, high signal-to-noise ratio, and high momentum-transfer ($q$) resolution. The results have established a detailed picture of the transient coupling between molecular spin-state switching and the lattice dynamic response triggered by laser-induced internal stresses in the thin films upon photoexcitation.

sphere in the UED experiment is nearly flat at the reciprocal space origin (Fig. 1d). Therefore, UED offers the advantages of efficient scattering at high-order Bragg peaks, providing extensive $q$-range coverage and large number of Bragg peaks probed. The electron beam direction is parallel to the orthorhombic $Z$ crystallographic direction of the highly-oriented thin film, so the UED pattern solely contains $hkl$ reflections with $l = 0$. Figure 2b depicts the change of the static electron diffraction pattern corresponding to the structural changes associated with the thermally-induced SCO transition from the LT to the HT phase. The changes in low $q$ near the beam stop in Fig. 2b are masked due to the sample thermal expansion and the slow pointing drift of the electron beam, which has been corrected. In Fig. 2c, the transient changes in the diffraction peaks at $+2$ ps after photoexcitation are similar to the thermally-induced changes in the static diffraction patterns in Fig. 2b, providing structural evidence of photoinduced SCO from the LS to the HS states. The subtle differences between the transient changes at $+2$ ps and thermally-induced changes indicate differences in molecular structure and lattice volume between transient out-of-equilibrium state at $+2$ ps and the thermal equilibrium in thermally-induced SCO.

Figure 2d shows the relative changes in the intensity of selected Bragg peaks on the ultrafast timescale (from $-1$ to $+4$ ps). The temporal response of the Bragg peaks in the electron diffraction patterns can be described as comprising both a fast and a slow component. These two components contribute differently from one Bragg peak to another. For example, the 020 and 430 Bragg peak intensities exhibit markedly different kinetic behaviors. The 020 peak exhibits a rapid intensity increase followed by a gradual drop in intensity, while the 430 peak intensity undergoes a depletion on two timescales. An investigation of the Debye-Waller (DW) effect (Supplementary Note 1) shows that this contribution to the time-resolved intensity change in Bragg peaks at $+2$ ps is relatively small.

The UED experiment, however, lacks $q$-resolution due to its large uncorrelated beam divergence[48] and the relatively modest changes of the $a$ and $b$ unit-cell parameters upon the SCO. As shown in Fig. 2b, c, radial shifts in Bragg peak positions cannot be resolved at the current signal-to-noise level for neither the thermally-induced nor the photoinduced SCO processes. Furthermore, the structural orientation of crystallites in films of **1** and the high electron energy (MeV) restrict the access to Bragg peaks of the $hk0$ family as depicted in Fig. 1d. This limitation makes the UED measurement insensitive to the lattice dynamics along the $Z$ crystallographic direction (normal to the surface), which requires Bragg peaks with $l \neq 0$ indices. Note that these limitations are not inherent to UED, but are specific to the sample and measurement conditions used.

Ultrafast XRD measurement conducted at the FXE beamline[47] at EuXFEL is complementary to UED as it samples a different part of reciprocal space, and it exhibits enhanced $q$-resolution due to its better beam divergence at the sample position while maintaining 25 fs pulse duration[49] and high brightness. A typical static X-ray diffraction pattern of a film of **1** is shown in Fig. 3a within a $q$ range of 0.8–3.0 Å$^{-1}$, which contains fewer Bragg peaks within a narrower $q$-range compared to the UED experiment. X-ray pulses with a photon energy of 12 keV probe several Bragg peaks with indices $l \neq 0$ due to the smaller curvature of the Ewald sphere and the broadening of the reciprocal lattice points (arising from the finite and imperfect nature of the thin-film sample) (Fig. 1d). Consequently, the XRD measurement allows for a detailed investigation of the lattice dynamics along the three $X$, $Y$, and $Z$ crystallographic directions, corresponding to $a$, $b$, and $c$ unit-cell parameters, employing the same geometry and photoexcitation conditions as in the UED measurement (Methods and Supplementary Note 2). Figure 3b, c show time traces of relative changes in intensity and radial position of selected Bragg peaks for the short timescale (from $-0.5$ ps to $+6$ ps).

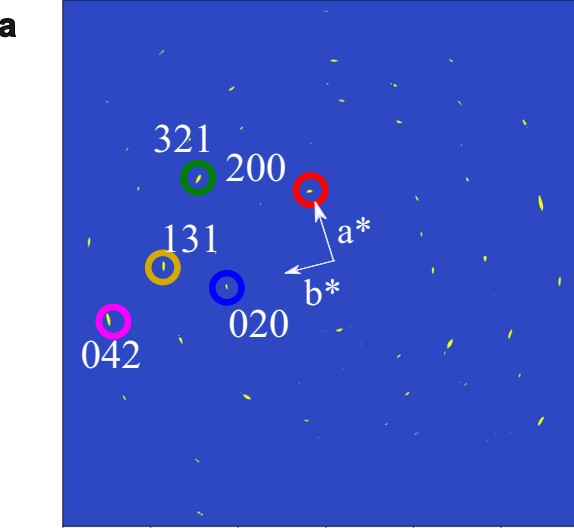

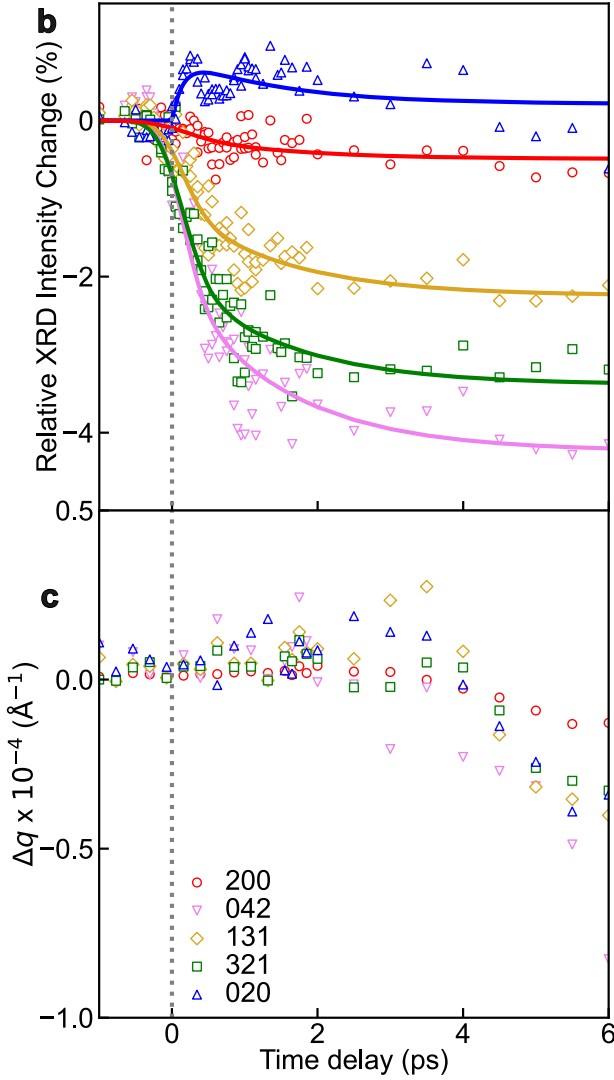

**Fig. 3 | Kinetic traces of selected Bragg peaks in ultrafast X-ray diffraction study. a** Static X-ray diffraction pattern measured at 300 K after scattering background correction. Color circles show the positions of selected Bragg peaks. **b** Kinetic traces of intensities of selected Bragg peaks from $-0.5$ to $+6$ ps. Solid line show the fitting curves using biexponential decay functions (Supplementary Eq. 2) (**c**) Kinetic traces of radial positions of selected Bragg peaks from $-0.5$ to $+6$ ps.

In Fig. 3b, the relative intensity changes in Bragg peaks in the XRD measurement show very similar biexponential dynamics to those observed in the UED measurement in Fig. 2d. For example, the intensity of the 020 Bragg peak consistently exhibits a rise and a decay component. Indeed, the kinetic traces of the Bragg peaks in both UED and XRD measurements can be globally fitted to a biexponential function. In the UED data, the first component is $147 \pm 44$ fs and the second is $1.44 \pm 0.78$ ps (Supplementary Note 3). In the XRD data, the timescales for the short relaxation process ($148 \pm 45$ fs) and the slower second component ($1.48 \pm 0.11$ ps) match those found from the UED data. In Supplementary Fig. 14, the transient changes in the XRD diffraction peaks at $+2$ ps after photoexcitation are similar to the thermally-induced changes in the static XRD diffraction patterns, like what observed in UED measurement (Fig. 2b, c). These results confirm that both UED and XRD measurements probe identical structural processes. It is worth mentioning that the excitation conditions used in the present work are within the linear single-photon regime (Supplementary Note 2), and the estimated low excitation fractions in both measurements ($\Delta X_{HS} \approx 3$ % and 6 %, in the UED and XRD experiments respectively; see Supplementary Note 6) are consistent with those in previous studies of photoinduced SCO in solids[17,19,20]. Figures 2d and 3b reveal some oscillations in Bragg peak intensities. However, due to the current limited signal-to-noise ratio for these weak oscillations, a solid interpretation of this dynamic behavior cannot be established, despite thorough analysis and comparison with existing literature[18,30] (Supplementary Discussion 1). Future studies with enhanced signal-to-noise ratios will focus on investigating these oscillations in greater detail.

Furthermore, in Fig. 3c, the radial position of Bragg peaks traces the lattice dynamic response along all three crystallographic directions, since the inter-planar distance for the orthorhombic system is given by $\frac{1}{d^2} = \frac{h^2}{a^2} + \frac{k^2}{b^2} + \frac{l^2}{c^2}$. No clear shift in peak positions can be noticed in the first 2 ps upon photoexcitation, indicating the absence of any unit-cell parameter changes on this short timescale. However, after $+2$ ps, a distinct radial shift in the position of Bragg peaks toward lower $q$ values begins to appear, indicating that lattice volume changes start to arise.

To reveal the molecular motions in the first 2 ps upon photoexcitation we used a parameterized molecular model[4,19–21,50]. For this model, we used most of the Bragg peaks observed in the UED and XRD results, and the intensity changes observed in 80 Bragg peaks from the UED results, along with 32 Bragg peaks from the XRD results were incorporated. Here, the complementary nature[48,51] of UED and XRD primarily lies in terms of $q$-resolution, $q$-range, and number of Bragg peaks in different parts of reciprocal space probed. The extensive $q$-range and the large number of Bragg peaks in the $hk0$ family from the UED results improve the spatial resolution of the molecular model and reduce the risks of overfitting that would arise if relying only on the limited number of Bragg peaks in the narrow $q$-range probed by XRD. In contrast, the Bragg peaks in different $hkl$ ($l=1$ or 2) family from the XRD results allows us to identify inconsistencies or artifacts in the data, improving the reliability and accuracy of the model (Supplementary Note 4). The experimental data from UED and XRD provides a robust foundation for modeling and allows us to confidently assign molecular structural dynamics. Additionally, the high $q$-resolution XRD results reveal a crucial detail: the structural dynamics of photoexcited molecules occur within a constant LT unit-cell volume during the first 2 ps.

The number of degrees of freedom in the model is kept to a minimum by carefully selecting key structural modes to determine the optimal global structure (Supplementary Note 4). This approach is based on the symmetric molecular structure of **1**, theory calculations on low-frequency vibrational modes, and studies of the reaction coordinates and structural dynamics of similar SCO systems[18–20,25,30]. An important aspect is that both UED and XRD probe the ensemble average of all interatomic distance changes within the unit cell. The observed evolution of the Bragg peak intensities reflects correlated

atomic motions driven by the photoinduced dynamics. Two independent dynamics groups were defined to decompose atomic motions from the LS ground state to the HS state into a linear combination of two reaction coordinates: symmetric Fe–ligand elongation with rigid ligands ($\mathbf{p}_{ELO}$) and ligand rotation toward fully-relaxed HS structure ($\mathbf{p}_{ROT}$) (Fig. 4a). The symmetric elongation of the Fe–ligand bonds, corresponding to the molecular breathing mode (Supplementary Movie 1), is the direct structural fingerprint of the SCO of **1** due to the

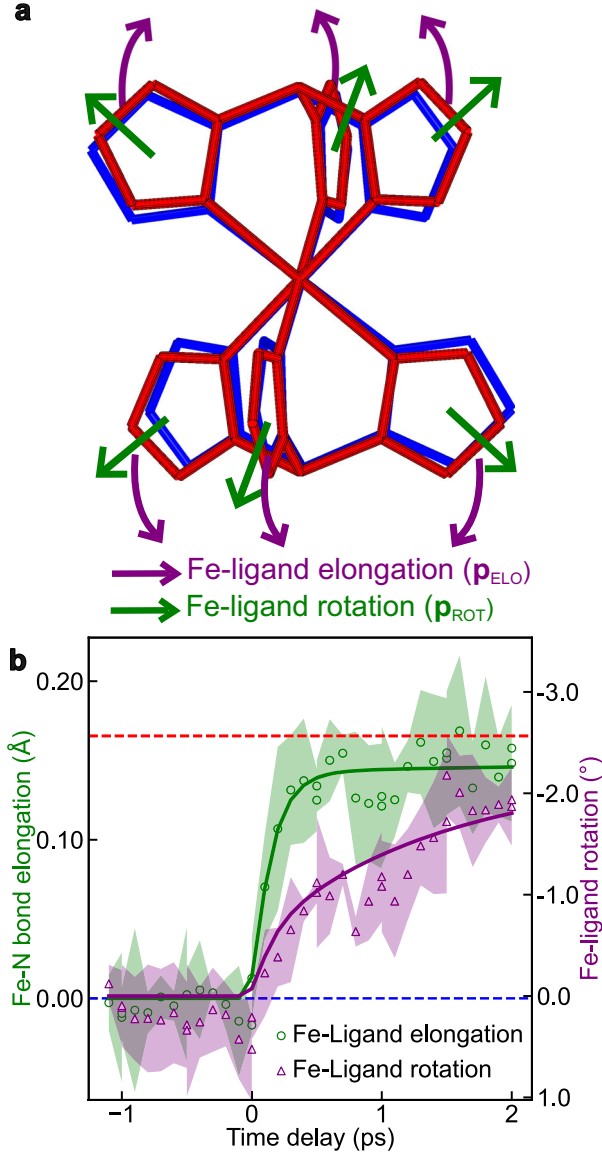

**Fig. 4 | Structural dynamics groups and temporal evolution. a** Structures of the molecule in the LS state (in the LT unit cell) (blue smooth lines) and in the HS state (converted into the same LT unit cell) (red fluted lines). The two structures overlap by their iron centers to better display the molecular structural changes occurring upon SCO. The atomic motions in **1** are decomposed into two structural dynamics groups schematized by arrows: symmetric Fe–ligand elongation with the rigid ligands ($\mathbf{p}_{ELO}$) and ligand rotation toward the HS structure ($\mathbf{p}_{ROT}$), **b** Temporal evolution of the Fe–ligand distance (average length of the Fe–N coordination bonds, green open circles) and of the ligand rotation (average N–Fe–N angles, purple open circles) in the ultrafast timescale (from −1 to $+2$ ps). Solid lines show the fitting curves using multi-exponential decay functions (Supplementary Eq. 2). Horizontal dashed lines correspond to the structural parameters in the HT phase (373 K) and LT phase (300 K), respectively. Error bars represent the full-width at half-maximum (FWHM) of the global maximum peaks of the Pearson correlation coefficient in reaction coordinate space (Supplementary Fig. 5).

population of antibonding orbitals of the iron center in the HS state. It has been widely used as the primary reaction coordinate describing structural change between the LS and HS states[18–20,25,30]. Molecular vibrational frequencies were computed for **1** in the HS state following geometry optimization (Methods and Supplementary Data 1). Among vibrational modes between 70 cm$^{-1}$ and 200 cm$^{-1}$, the frequency of the breathing mode (136.9 cm$^{-1}$) is similar to those of other similar SCO systems[18,20,30,31]. The ligand rotation has the effects of changing the N–Fe–N angles without significantly modifying the Fe–ligand bond distances (Supplementary Movie 2). This rotation movement is triggered by the Fe–ligand elongation due to the rigidity of the ligands[25,30]. Other low-frequency modes, such as ligand torsion (Supplementary Movie 3) and out of phase Fe–ligand stretching (Supplementary Movie 4), are reported to be irrelevant during vibrational cooling[25,30,31].

The parameterized molecular model determines the best fit structure of **1** at each time delay by maximizing the correlation between the experimental and simulated diffraction intensities (comprising 112 Bragg peaks in total from both UED and XRD experiments), covering the full parameter space between the LS and HS structures (Supplementary Fig. 4). In this way, we derive the temporal evolutions of the Fe–ligand distance (average length of Fe–N coordination bonds) and of the ligand rotation (average N–Fe–N angles), as depicted in Fig. 4b. These two key structural modes can be related to the fast and slow components observed in the UED and XRD data.

The 150-fs short Fe–ligand elongation process is compatible with the ultrafast ISC processes observed in Fe$^{II}$ complexes involving the population of antibonding $e_g^*$ orbitals[18,52–54] (taking into account the 100-fs FWHM instrument response function of the UED and XRD setups). The second approximately 1.45-ps ligand rotation is accompanied by structural relaxation of the photoexcited (vibrationally hot) molecules in the HS potential well leading to the fully relaxed HS excited state structure[14,17,18,22–24,31]. During this relaxation phase, energy is redistributed to the lattice and other low-frequency modes. In Fig. 4b, following the IVR process at +2 ps, the measured Fe–ligand elongation is comparable to that observed for HS molecules in HT unit cell, while the ligand rotation is found to be smaller. The structure of the photoinduced HS state at +2 ps thus differs to some extent from that found in the thermally-induced SCO due to the internal chemical pressure[55] from neighboring unrelaxed unit cells, which restrains the structural relaxation of the photoexcited molecules.

Our experimental data, with high temporal and $q$-resolution, unambiguously reveal that the local structural rearrangement of photoexcited molecules takes place within a constant LT unit-cell volume on an ultrafast timescale (<2 ps). This critical observation supports our finding of a non-equilibrium state, where the structural relaxation of photoexcited molecules is constrained by the chemical pressure exerted by neighboring unit cells in Fig. 4. We further confirm the photoexcited spin dynamics in SCO systems through the direct observation of nuclear rearrangements, combined with electronic dynamics from previous studies that allows for the assignment of the spin states involved[14,17,18,22–24,31,52–54]. Notably, our results provide direct insights into how the sequence of atomic motions–Fe–ligand elongation followed by ligand rotation–stabilizes the HS state at the earliest femtosecond timescales. These two atomic motions were previously reported as coupled motions due to the (0.4 ± 0.05)-ps instrument response time of keV UED[20], whereas their sequential nature had been suggested by indirect spectroscopic studies[25,30,31]. Our diffraction data captured the atomic motions of entire molecules, in contrast with previous studies[25,30,31] that resolved specific vibrational modes and X-ray absorption bands[18,30], which mainly focused on changes in metal–ligand distances. The observed sequential atomic motions localized the correlated nuclear motions corresponding to the electronic dynamics on the Fe$^{II}$ metal center, from the initial LS to the HS potential surface[20,30,31]. The initial Fe–ligand elongation, driven directly by the change in electron configuration, triggers the

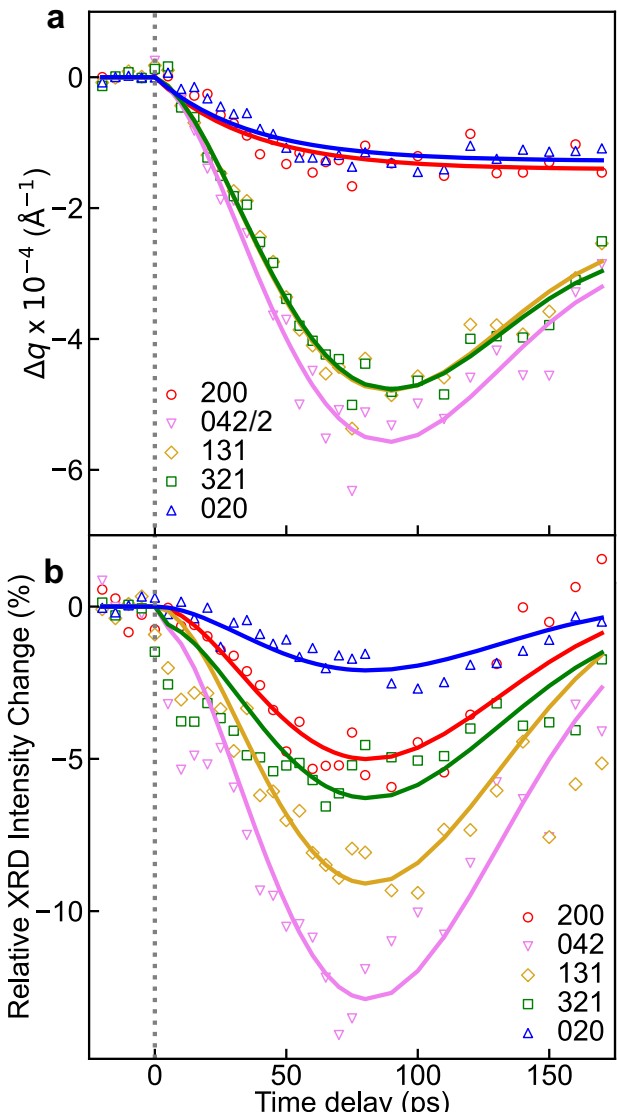

**Fig. 5 | Kinetic traces of selected Bragg peaks in long timescale. a** Kinetic traces of radial position of selected Bragg peaks from −10 to +175 ps. The solid lines for Bragg peaks with $l \neq 0$ show the results of the fitting analysis (Supplementary Eq. 2). The solid lines of $hk0$ Bragg peaks show monoexponential decay fits (Supplementary Eq. 2). The radial position shift of the 042 Bragg peak was scaled with 0.5 to emphasize the relatively small intensity changes for the 200 and 020 Bragg peaks. **b** Kinetic traces of intensities of selected Bragg peaks from −10 to +175 ps. Solid lines show results of the fitting analysis (Supplementary Eq. 2).

subsequent ligand rotation due to the ligand rigidity. This rotation is identified as a key mode activated to stabilize the photoexcited molecules during the vibrational cooling, extending beyond the traditionally-studied reaction coordinate of Fe–ligand distances[18,31,53,56]. As shown in Fig. 3d, after +2 ps, a subsequent shift in the radial position of Bragg peaks toward lower $q$ range is observed, signaling a lattice volume expansion. As discussed later, this global expansion of the unit-cell volume on the picosecond timescale mainly arises from the lattice temperature change as a large quantity of energy is transferred to the lattice during vibrational cooling (*vide infra*)[19].

## Dynamic lattice response and interaction between spin-state switching and photothermal heating

Figure 5a, b cover time traces of relative changes in radial position and intensity of selected X-ray Bragg peaks for the long timescale (from −25 to +175 ps). Figure 6a–e present the time-resolved differences of

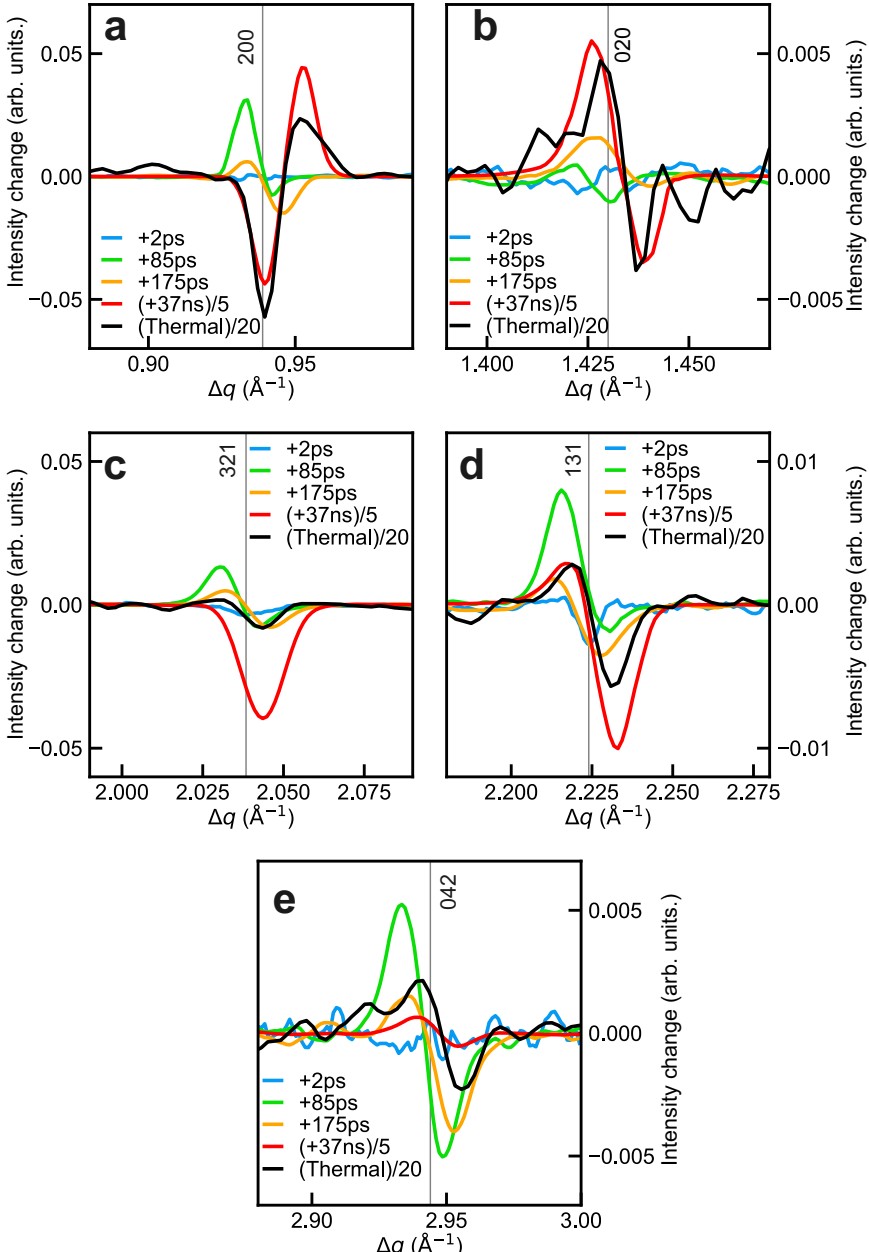

**Fig. 6 | Kinetic differences in X-ray diffraction study of selected Bragg peaks measured at 300 K. a–e** Photoinduced difference at selected time delays for the 200, 020, 321, 131, 042 Bragg peaks. The selected time delays are +2 ps (blue), +85 ps (green), +175 ps (orange) and +37 ns (red). The +37 ns photoinduced difference is scaled (by a factor of 5) to show the smaller changes in the early time delays. Scaled thermally-induced differences between the 313 K and 353 K phases are also plotted (black), which are ± 20 K to the transition temperature of 336 K. Note that these thermal differences represent a total change of HS fraction of 85%. The vertical lines show the Bragg peak positions at 300 K.

200, 020, 321, 131, and 042 Bragg peaks at selected time delays from +2 ps to +37 ns after photoexcitation. The thermally-induced differences of these Bragg peaks between 313 and 353 K are also presented.

As shown in Fig. 5a, the radial position of selected Bragg peaks with $l \neq 0$ exhibits a shift toward lower $q$ values accompanied by an oscillation. Note that such oscillations in the time evolution of $00l$ Bragg peak positions in thin films of **1** were also reported in a recent study[15]. On the other hand, the radial positions of the 200 and 020 Bragg peaks (with $l = 0$) in Fig. 5a undergo a monoexponential decay (Supplementary Eq. 2) toward lower $q$, with respective time constants of $38 \pm 7$ ps and $32 \pm 6$ ps, then remaining constant over subsequent timescales. In Fig. 5b, the relative changes in intensity of selected Bragg peaks for the long timescale show a very similar oscillation as observed in Fig. 5a. Specifically, concerning the 020 Bragg peak, the dynamics

consist of a rapid increase in intensity on the sub-ps timescale due to ultrafast molecular switching (Fig. 3b), followed by an oscillating decrease in intensity on the tens of ps timescale (Fig. 5b).

While the sample was excited in the MLCT band in single-photon regime (Methods), it is noteworthy that the photon energy (267 nm, 4.64 eV) is much larger than the energy difference between the LS and HS states (≈100 meV). The relaxation of the photoexcited, vibrationally hot HS molecules occurs via IVR and vibrational cooling through energy exchange with the surrounding lattice. As a result, a large quantity of energy is transferred to the lattice, involving the generation of photoinduced stresses in the crystal lattice. The mechanical equilibrium with the environment is then restored via the coherent propagation of acoustic strain waves within the sample, which results in an expansion of the lattice volume. The distinct behaviors observed in

Bragg peak positions with $l \neq 0$ and $l = 0$ in Fig. 5a show that the lattice volume expansion occurs with different dynamics in and out of the sample plane.

We globally fitted the oscillations shown in Fig. 5a, b (Supplementary Eq. 2), resulting in a period of oscillations ($T_{osc} = 221 \pm 15$ ps). As the strain impedance of the film is smaller than that of the substrate, it takes two round trips for the propagating strain to produce constructive interferences ($T_{osc} = 4L/v$)[15]. Given the film thickness ($L = 200 \pm 10$ nm), our fitting results with $v = 3.6 \pm 0.3$ m s$^{-1}$ can qualitatively reproduce the observed results in Fig. 5a, b. This value of $v$ is in line with the longitudinal speed of sound ($v = 3.0 \pm 0.3$ km s$^{-1}$) previously reported[15]. Therefore, both oscillations in radial position shifts and intensity changes for Bragg peaks with $l \neq 0$ are associated with the acoustic resonance of thin films[57] along the $Z$ crystallographic direction. These typically arise from an acoustic impedance mismatch, generating back reflecting strain waves at interfaces within the film[15]. These acoustic phonons modulate the spacing between lattice planes so that the oscillation in radial position of the Bragg peaks reflects a periodic compression and expansion of the lattice. The oscillations in Bragg peak intensity can be attributed to constructive and destructive interferences in diffraction induced by periodic displacements of atoms in the lattice associated with the breathing acoustic propagation along the $Z$ crystallographic direction[9,19].

For the monoexponential decay of the radial positions of the 200 and 020 Bragg peaks toward lower $q$, interfacial impedance mismatch may arise at the borders of laser irradiated surface, or domain boundary in the $XY$ plane. Strain waves propagate at transverse acoustic velocity over distances of a few hundred micrometers of the laser beam waist unimpeded by interferences, hence no oscillations are observed in the $XY$ plane. Therefore, the radial position shifts of 200 and 020 Bragg peaks, which are insensitive to propagating strain waves along $Z$ crystallographic direction, report the time-resolved lattice dynamical response along the $X$ and $Y$ crystallographic directions. The time constant of the monoexponential decay dynamics shown in Fig. 5a closely aligns with the longitudinal acoustic timescale[6,58], which can be estimated as $L/v \approx 50$ ps for the film analyzed in this study. Our present XFEL results, containing a better temporal resolution than synchrotron scattering experiments[14,15], demonstrate that the strain wave dictates the pathway of a long-range volume expansion in the three crystallographic directions. These strain waves also induce changes in the structure factor by modulating the overall atomic positions, leading to observable intensity oscillations for Bragg peaks even with $l = 0$.

It is important to note that, as mentioned above, the specific evolution of the $a$ unit-cell parameter of **1** offers the opportunity in the present study to disentangle the time-resolved structural changes in the crystal lattice that arise purely from thermal effects and those that are caused by the SCO switching process. Indeed, distinguishing between these two contributions to the changes in the unit-cell volume remains a challenging task[14,19,20,29].

Interestingly, our results in tens of ps timescale reveal that both 200 and 020 Bragg peaks display a radial position shift towards lower $q$ in Figs. 5a and 6a, b. However, a different behavior is expected for these two Bragg peaks upon the LS-to-HS molecular switching (shrinkage along $X$ crystallographic direction, but elongation along $Y$ crystallographic direction)[15,44]. The fact that both 200 and 020 Bragg peaks exhibit a similar position shift towards low $q$ values shows that, in the present experimental conditions (UV light excitation), the unit-cell volume expansion on this timescale mainly arises from lattice heating (ordinary thermal expansion). An upper limit for the lattice temperature rise of $\Delta T_{latt} = 60$ K can be estimated from the deposited laser power, assuming complete photon energy conversion to heat and neglecting radiative losses (Supplementary Note 5). From the experimentally observed lattice expansions at +100 ps (Fig. 5a) and the known thermal expansion coefficients along the crystallographic axes[15], a temperature increase of about 40 K can be estimated

**Table 1 | Changes of unit-cell parameters at different time delays**

| | Changes of unit-cell parameters [× 10$^{-3}$ Å] | | | | |
|---|---|---|---|---|---|
| | + 2 ps | + 85 ps | + 175 ps | + 37 ns | thermal SCO |
| $a$ | 0 | + 14 | + 16 | − 369 | − 233 |
| $b$ | 0 | + 6 | + 9 | + 74 | + 84 |
| $c$ | 0 | + 106 | + 74 | + 452 | + 975 |

(Supplementary Note 5), which matches the value obtained from the laser fluence. It is worth mentioning that while the estimated $\Delta T_{latt}$ could bring a global lattice temperature above the spin transition temperature $T_C$, in Fig. 5b there are no discernible changes in intensity of Bragg peaks associated with further molecular switching (i.e., no substantial increase in fractional population of photoexcited HS molecules ($\Delta X_{HS}$)), other than the oscillations due to propagating strain waves. This observation aligns with findings from complementary transient absorption spectroscopy and XRD studies of films of **1**[15,16], showing a constant $\Delta X_{HS}$ on this sub-nanosecond time domain after the initial photoinduced SCO.

On the other hand, on the tens of ns timescale, substantial changes in intensity and position of Bragg peaks are observed in the difference pattern at +37 ns (Fig. 6a–e). In particular, our results reveal that the radial position of the 200 Bragg peak (Fig. 6a), which was initially shifted towards lower $q$ at $t = 175$ ps (compatible with ordinary thermal expansion), is found to be shifted towards higher $q$ values at $t = +37$ ns (meaning a decrease of the $a$ unit-cell parameter). Therefore, along with the above-mentioned specific evolution of the unit-cell parameters of **1**[15,44], this drastic change in the position shift of the 200 peak clearly indicates an additional switching of a large fraction of molecules into the HS state. This so called thermo-elastic SCO step, occurring in the ns timescale, is delayed compared to the lattice expansion due to the existence of an energy barrier between the LS and HS states at the molecular scale, as discussed in previous studies[14–16]. In addition, other Bragg peaks, including 020, 131, and 042 reflections, display a further position shift towards lower $q$ at +37 ns, indicating an additional increase of the $b$ and $c$ unit-cell parameters compared to $t = +175$ ps, which is compatible with a noticeable thermo-elastic SCO step. The changes in unit-cell parameters at different time delays are summarized in Table 1. In Fig. 6, the deviations between the photoinduced difference at +37 ns (red) and the thermally-induced difference (black) are thought to arise from the mechanical stresses expected in the out-of-equilibrium state at +37 ns, due to an inhomogeneous distribution of unit cell volumes (Supplementary Fig. 9). As estimated from the structure factor of Bragg peaks, the fraction of photoswitched HS molecules is found to reach 53 % at +37 ns (Supplementary Note 6). This delayed thermo-elastic SCO step occurs since the fraction of HS molecules equilibrates with the new lattice conformation, predominantly due to the transient increase of the lattice temperature. On the thermal spin-transition curve[15] at equilibrium, the estimated value of $\Delta X_{HS}$ (53 %) corresponds to a temperature increase of approximately 40 K from room temperature. This aligns well with the observed increases of the $a$ and $b$ unit-cell parameters at +100 ps and with the value of $\Delta T_{latt}$ estimated from the laser fluence (60 K), reaffirming the thermally activated nature of this delayed switching step.

Our present study of the dynamic lattice response shows that although both molecular switching and lattice heating are known to be responsible for lattice volume changes in thin films of **1** at equilibrium, their respective contributions are different in different timescales after photoexcitation, as revealed in Figs. 5 and 6. The experimental results suggest that during the ultrafast timescale (< 2 ps), the photoinduced molecular switching does not trigger any lattice dynamics, and no significant lattice heating is observed during this period. Subsequent to ultrafast photoswitching, the lattice heating becomes dominant in

terms of lattice volume change on the timescale of several to hundreds of picoseconds, owing to the high photon energy required for MLCT excitation. However, as a consequence of the energy barrier existing between the LS and HS states at the molecular level[14–16,25], a significant proportion of LS molecules is switched to the HS state only over nanoseconds (during the thermo-elastic SCO step), so that the main contribution to the lattice deformations at this timescale becomes the substantial thermally activated molecular switching.

Additionally, it is worth mentioning in our work that obtaining a fraction of photoswitched molecules as high as 53 % using a single fs laser pulse in a reversible fashion at 10 Hz is particularly noteworthy for potential applications of photoswitches and photo-functional materials. In previous studies, the maximum reversible photo-induced switching fraction was restricted to 7–8% in the same film (using different excitation conditions)[16] and 25% in another SCO film[13,14]. Here, our findings demonstrate the ability to switch a substantial fraction of molecules below sample damage threshold using an ultrafast laser, which is crucial for achieving significant and rapid modulation of various physical properties.

We investigated a prototypical thin film of the SCO complex [Fe(HB(tz)$_3$)$_2$] exhibiting structural dynamics consistent with an abrupt spin transition near room temperature and distinct lattice response during laser-induced molecular switching and ordinary thermal expansion. Owing to the highly oriented structure of the film, it provides a foundation for controlled and systematic studies, enabling homogenous photoexcitation and crystallographic studies of the time-resolved dynamics from the molecular to the material scale. Our results, in good agreement with previous studies[13–16,20,30], provide insights to draw a comprehensive picture of photo-induced SCO dynamics down to the earliest fs timescale. In particular, the molecular motions in the first 2 ps upon photoexcitation revealed by our data disclose a curved trajectory in the space defined by sequential Fe−ligand elongation and ligand rotation, which operate in the photo-conversion process of molecules from the LS to the HS state on the sub-ps timescale, while the unit-cell volume remains constant (Fig. 7a).

Beyond this local structural rearrangement of the photoexcited HS molecules, our measurements also reveal the subsequent bulk lattice volume changes over time, as both the molecular structural changes and unit-cell deformations were simultaneously probed in out-of-equilibrium conditions with high structural sensitivity. Interestingly, different lattice dynamics are evidenced in and out of the sample plane. While acoustic oscillations are observed along the Z crystallographic direction (normal to the sample plane) due to the propagation of strain waves back reflected at interfaces, a mono-exponential expansion of the lattice is evidenced in the XY plane of the SCO film. As illustrated in Fig. 7b, a temporal separation is observed between the lattice volume expansion triggered by propagating strain waves and the subsequent delayed thermo-elastic molecular switching in the expanded and hot crystal lattice.

Our study also extends the understanding of lattice dynamics and origins of photo-induced elastic stresses at the most fundamental level. Indeed, due to the unique structural features of the investigated compound, our data allowed us to distinguish the respective contributions of lattice heating and molecular switching in the bulk unit-cell deformations. Although lattice heating is the primary effect on a short time-scale due to the energetic (UV light) excitation conditions, the significant lattice volume deformations observed around ten nanoseconds appear to be mainly due to the large molecular rearrangement associated with the LS-to-HS transition. Undoubtedly, these structure-specific interactions, resulting from structural rearrangement in photo-excited molecules and long-range lattice volume changes in the bulk material, constitute an interesting scope for future investigation of photoinduced phase transition in molecular thin films amenable to applications.

Furthermore, this study highlights opportunities provided by modern ultrafast sources, such as MeV-electrons and XFELs, in

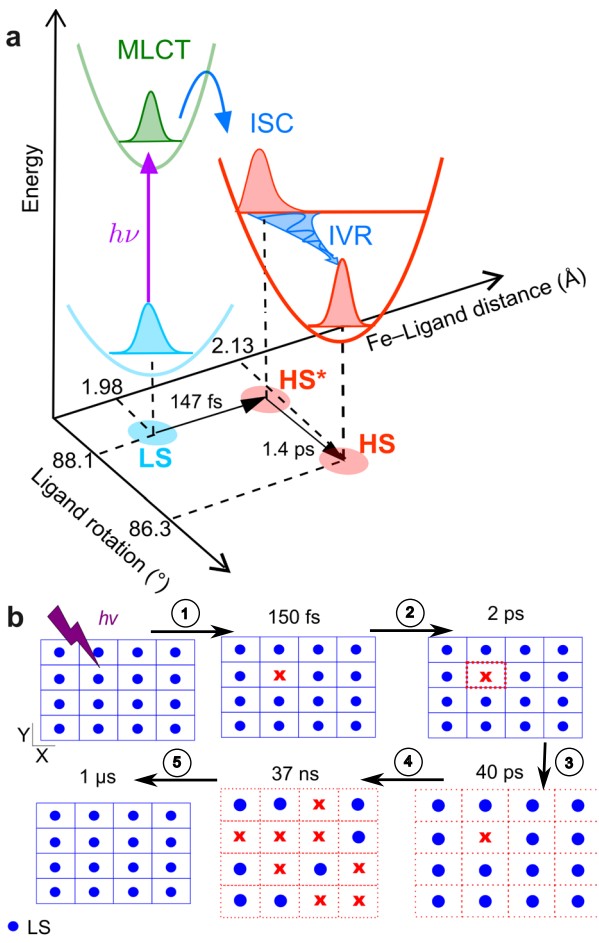

**Fig. 7 | Schematic of the proposed pathway of the photoinduced phase transition revealed by the UED and XRD experiments. a** A curved trajectory in the space defined by Fe−ligand elongation and ligand rotation presents a comprehensive picture of the sequential photo-switching dynamics down to the earliest fs timescale. Upon photoexcitation from the LS ground state, the molecule reaches the HS potential surface associated with an ultrafast Fe-ligand elongation through ISC processes. IVR activates ligand rotation, promoting further reorganization of the excited HS state (HS*) within the constant LT unit cell. **b** The scheme of photo-excitation cycle. 1: the pump laser excites a small fraction of the LS molecules (blue dots) in the LS lattice (blue grids) into hot HS molecules (red crosses) in less than 150 fs. 2: molecular vibrational cooling process happens in the constrained unit cell in a few picoseconds, transferring a great quantity of energy to the lattice. 3: As a result, the unit-cell volume expands with the propagation of strain wave in ~40 ps from blue solid lines to red dash lines, indicating a temporal decoupling between the volume expansion and molecular switching. 4: A second significant SCO step occurs in tens of nanosecond since the fraction of LS/HS molecules equilibrates with the new lattice conformation with expanded volume and elevated temperature. 5: The excited sample relaxes back to the initial ground state in 1 μs timescale[15,16].

providing a deeper understanding of photoinduced structural dynamics[48,51]. The combined UED and XRD measurements, with different q-resolution, q-coverage, and the number of Bragg peaks from different crystal planes probed in this work, provide a robust foundation for structural refinement and allows for a more complete understanding of the ultrafast lattice dynamics in SCO materials.

## Methods
### Sample preparation and characterization
The Fe(HB(tz)$_3$)$_2$ (tz = 1,2,4-triazol-1-yl) thin film was prepared using the same method as reported previously[38]. The molecules were deposited by vacuum thermal evaporation method on 50-nm-thick silicon nitride membranes.

Given the absorption coefficient of **1** at 267 nm ($\alpha_{267nm} =$ 45400 cm$^{-1}$), a film thickness of $200 \pm 20$ nm was selected to approximately match the laser penetration depth ($\delta_{267nm} = 220$ nm), enabling rather homogeneous excitation. The SCO film consists in highly oriented crystallites growing with the $Z$ crystallographic direction normal to the surface of the film[38]. As shown in Fig. 1c, diffraction measurements on thin films of **1** from UED and XRD show single-crystal-like patterns, allowing us to accurately determine the radial position of Bragg peaks. The temperature-dependent X-ray diffraction study was performed on the thin film at the EMBL P14.EH2 (T-REXX) end-station of PETRA-III (DESY, Hamburg, Germany) with the Dectris Eiger 4 M detector. The sample was mounted in transmission geometry on a thermoelectric stage from Linkam Scientific. The temperature was set to 313 K for LS state and 353 K for HS state. These two temperatures correspond to 85 % of the transition in HS fraction[43]. The X-ray beam energy was set to 12.7 keV ($\lambda = 0.97622$ Å) containing $4.64 \times 10^{12}$ photon s$^{-1}$ with an elliptical-shape beam (spot size of 50 (H) × 10 (V) μm) at the sample position. For different temperatures, patterns were collected with 10% transmission with 50 ms exposure time. The measurement was performed on six different sample positions.

## Laser excitation conditions

The two experiments (UED and XRD) were conducted under similar excitation conditions. At the sample position, the pulse duration of the pump laser was 70 fs, and the wavelength was centered at 267 nm. Photoexcitation of molecules in the LS state at this wavelength induces a metal-to-ligand-charge-transfer (MLCT) transition[16,59]. The incident excitation fluence was 1.47 mJ cm$^{-2}$ or 20.93 GW cm$^{-2}$ for UED and 2.76 mJ cm$^{-2}$ or 39.38 GW cm$^{-2}$ for XRD, respectively. The fluences used in both UED and XRD were below the sample damage threshold and in the linear range of the excitation (Supplementary Note 2). The excitation fraction (fractional population of photoexcited HS molecules) was calculated by two independent methods, based on the incident laser fluences (knowing the optical absorption properties of **1**) and from the relative intensity changes in Bragg peaks. (Supplementary Note 6).

## UED experiment

The thin-film sample was measured by UED in transmission geometry with 3.5 MeV ($\lambda = 3.6 \times 10^{-3}$ Å) electron beams available at the Shanghai Jiao Tong University (Shanghai, China)[45]. The sample was oriented so that the electron beam was parallel to the $Z$ crystallographic direction. The orientation was indexed to be parallel to the (110) crystal plane (Fig. 2a). Single electron pulse contained $1.2 \times 10^4$ electrons with a circular-shape beam (spot size of ($270 \pm 10$) μm FWHM) at the sample position. The temporal resolution was 100 fs FWHM to increase brightness of electron pulses for higher signal-to-noise ratio. During the data collection, the same sample position was pumped by the laser and probed by electron pulses with a repetition rate of 100 Hz. This allows the excited sample to relax back to the initial ground state before the arrival of next pump pulse. The experiment was performed at 300 K.

## XRD experiment

The time-resolved XRD measurements of the thin-film sample were performed in the same transmission geometry at the FXE beamtime[46,47] at the EuXFEL (Schenefeld, Germany). The sample was mounted such that the X-ray pulses were perpendicular to the surface of the sample. The samples were maintained in the LS state at 300 K. The X-ray photon energy was 12 keV ($\lambda = 1.03317$ Å) with 0.25% bandwidth. During one set of pump-probe delay scans, the same sample position was pumped and probed as the UED experiment. Due to the slow and dose-dependent X-ray-induced damage, a new sample position was used for each set of data collection. A single X-ray pulse contained $1.56 \times 10^{10}$ photons. The X-ray beam had a roughly elliptical shape, and its focus

size was $60 \times 40$ μm$^2$ FWHM at the sample position. The X-ray pulses were attenuated to avoid saturation on the detector. The temporal resolution was 115 fs FWHM[47]. The measurement was performed with 10 Hz repetition at 300 K. The 1 M Large Pixel Detector (LPD)[60,61], a hybrid pixel X-ray detector developed by the Science and Technology Facilities Council (STFC, United Kingdom), was used for the data collection. Data reduction was performed by a fast azimuthal integration algorithm using the PyFAI Python library[62].

## Quantum chemistry calculations

The geometry optimization and vibrational frequencies of the HS state of **1** were calculated using the B3LYP functional[63,64] and the LANL2DZ basis set included in the Gaussian 16 program package[65]. The atomic coordinates in the HS state were taken from the literature[66]. The calculated result is provided in Supplementary Data 1.

## Data availability

The reduced experimental data that supports the findings of this study are available within the article and its Supplementary Information files. Any additional information will be made available by the corresponding authors upon request. Source data are provided with this paper.

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

## Acknowledgements

This research was supported by the R&D project "XTRAS" in EuXFEL (D.V. and Y.J.), and received financial support from the Agence Nationale de la Recherche (Project SCOPOL, ANR-22-CE09-0019) (K.R.), (Project MULTICROS, ANR-19-CE29-0018) (M.L.). We are grateful to L. Zhang for help with the film deposition, and to T. Renaud for much appreciated help during beamtime. D.X. acknowledges support from the National Natural Science Foundation of China (grants No. 11925505 and 12335010) and from the New Cornerstone Science Foundation through the XPLORER PRIZE. The UED experiment was supported by the Shanghai soft X-ray free-electron laser facility. P.Ł., R.K., and K.N.J. thank the National Science Centre (Poland) (SONATA: grant No. 2016/21/D/ST4/03753, and SONATA BIS: grant No. 2020/38/E/ST4/00400, programs) and Ministry of Science and Higher Education (Poland) (agreement No. 2022/WK/13) for financial support. R.M.V. acknowledges funding by the Initiative and Networking Fund of the Helmholtz Association. R.W. acknowledges funding from the German Academic Exchange Service (DAAD). H.M. acknowledges generous funding by the Deutsche Forschungsgemeinschaft (DFG, German Research Foundation) under Germany's Excellence Strategy (EXC 2008/1 – 390540038, UniSysCat) and within SFB/CRC 1636, ID 510943930 - project A01.

## Author contributions

Y.J., K.R., R.M.V., G.M., C.M., and M.L. conceived the experiments. K.R., G.M. performed the sample preparation. Y.J., K.R., R.S., C.D., N.B., T.S., and H.M. performed the sample characterization with TEM and spectral measurements. Y.J., L.L., F.Q., and D.X. performed the ultrafast electron diffraction experiment, analyzed, and interpreted the data with D.V., K.R., C.M., and M.L., Y.J., D.V., and D.v.S. performed the temperature-dependent X-ray diffraction measurement. Y.J., D.V., K.R., T.E., M.J., F.A.L., P.Z., D.K., C.D., H.Y., P. Ł., R.K., K.N.J., R.W., R.M.V., H.M., C.M., and M.L performed the X-ray diffraction experiment at the FXE beamline of EuXFEL, and interpreted the data with L.L. and G.M. All authors discussed results and contributed to the writing of the manuscript.

## Funding

## Competing interests

The authors declare no competing interests.

## Additional information

¹European XFEL, Holzkoppel 4, Schenefeld, Germany. ²Laboratoire de Chimie de Coordination, CNRS UPR 8241, Université de Toulouse, 205 route de Narbonne, Toulouse, France. ³Key Laboratory for Laser Plasmas (Ministry of Education), School of Physics and Astronomy, Shanghai Jiao Tong University, Shanghai, China. ⁴Collaborative Innovation Center of IFSA, Shanghai Jiao Tong University, Shanghai, China. ⁵Uncharted Software, 600-2 Berkeley St., Toronto, ON, Canada. ⁶Condensed Matter Physics (KOMET), Institute of Physics, Johannes Gutenberg University Mainz, Staudingerweg 7, Mainz, Germany. ⁷Center for Data and Computing in Natural Sciences (CDCS), Notkestrasse 10, Hamburg, Germany. ⁸Deutsches Elektronen-Synchrotron DESY, Notkestrasse 85, Hamburg, Germany. ⁹European Molecular Biology Laboratory (EMBL), Hamburg unit c/o DESY, Notkestr. 85, Hamburg, Germany. ¹⁰University of Warsaw, Faculty of Chemistry, Żwirki i Wigury 101, Warsaw, Poland. ¹¹Department of Chemistry, University of Illinois Urbana-Champaign, Urbana, IL, USA. ¹²Institute of Chemistry, University of Potsdam, Karl-Liebknecht-Str. 24-25, Potsdam-Golm, Germany. ¹³Helmholtz Zentrum Berlin für Materialien und Energie GmbH, Hahn-Meitner-Platz 1, Berlin, Germany. ¹⁴Institute of Optics and Atomic Physics, Technical University of Berlin, Berlin, Germany. ¹⁵Zhangjiang Institute for Advanced Study and Tsung-Dao Lee Institute, Shanghai Jiao Tong University, Shanghai, China. ¹⁶Univ. Rennes, CNRS, IPR (Institut de Physique de Rennes), UMR 6251, Rennes, France. ¹⁷CNRS, Univ Rennes, DYNACOM (Dynamical Control of Materials Laboratory), IRL2015, The University of Tokyo, 7-3-1 Hongo, Tokyo, Japan. ✉e-mail: dxiang@sjtu.edu.cn; christopher.milne@xfel.eu; maciej.lorenc@univ-rennes1.fr; yifeng.jiang@xfel.eu

