## [Transparent Peer Review file · Nature Communications]

Capturing ultrafast molecular motions and lattice dynamics in spin crossover film using femtosecond diffraction methods

Corresponding Author: Dr Yifeng Jiang

Version 0:

Reviewer comments:

Reviewer #1

(Remarks to the Author)

The manuscript entitled "Capturing ultrafast molecular motions and lattice dynamic response in a spin crossover thin film using femtosecond electron and X-ray diffraction" by Y. Jiang describes the complementary ultrafast time-resolved X-ray and electron diffraction experiments on spin crossover complex crystals. The manuscript is well-written, and the characterization of molecular crystals with ultrafast X-ray and electron diffraction is quite a hot topic. However, I have several questions before I support this manuscript to be published in Nature Communications.

1. On page 3, the authors claim that "However, the exact time evolution and interaction between molecular structural rearrangement, photothermal heating, change in the unit-cell volume, and subsequential spin-state switching in (nano-)crystals on ultrafast timescales remains a subject of ongoing debate." However, what are specific big 'questions' on this material or this class of materials?
2. The observations (the changes in 150 fs, 2 ps, 40 ps, 37 ns, and acoustic phonon vibration) correspond well to the previous reports, and the authors completely summarized the dynamics in this manuscript, which is excellent. Furthermore, the dynamics are observed with state-of-the-art X-ray free electron laser and MeV time-resolved electron diffraction setup, which is also very interesting. However, what is newly understood in this manuscript? I believe that the structural dynamics in 150 fs and 2 ps (so far only in optical methods) seem new, which is still unclear in the manuscript.
3. The authors used complementary methods with X-ray and electron diffraction measurements. However, they used the results from X-ray diffraction for most of the discussions. In principle, both the X-ray and electron diffraction measurements similarly observe the lattice or atomic dynamics. The results of the electron diffraction measurements are just used to assist the X-ray diffraction results. Are the results of the electron diffraction measurements needed to support the discussion in this manuscript? Are the electron diffraction and X-ray diffraction measurements complementary used in this manuscript?
4. The point is quite interesting: approximately 5% of molecules in the crystal are photoexcited in a few picoseconds, the energy is distributed to the surrounding molecules, and more than 60% of molecules in the crystal undergo phase transition. The energy is still at the localized molecular sites in 2 ps, will be spread out to the lattice in 40 ps, and changes the surrounding molecules to HS state in 37 ns. Do the molecules have energy in the thermal form or transient intermediate structural form in 40 ps? All the intensities of X-ray diffraction spots decrease around 100 ps, as shown in Fig. 5.

Minor points

1. Fig. 7b is difficult to follow. Please add an arrow for the eye guide or put the figures from left to right.
2. I would like to see the diffraction intensity changes of electron and X-ray diffraction measurements in the same index in the supplementary information.
3. In Supplementary Methods 3, there is no Fig. 2e in the main text.
4. In Supplementary Methods 6, which structure factors do the authors use? All the structure factors they measured?

Reviewer #2

(Remarks to the Author)

In this work, the author studied the ultrafast molecular motion of a spin crossover molecular thin film using both MeV-UED and XFEL, two state-of-the-art femtosecond diffraction techniques. They have observed the photoinduced local SCO from LS to HS on sub-picosecond timescale, followed by a heating process on the 100 ps timescale, then followed by a global SCO on tens of ns timescale. They successfully combined the advantages of two distinct diffraction methods, and built a thorough picture of the structural dynamics across a wide range of temporal scale. I think the scientific merit for sure meets the standard of Nature Communication. However, I think the current version of the manuscript still need to be further clarified / improved.

Major points:

1. In the previous synchrotron scattering experiment (ref. 34), it is already established that heating occurs on 100 ps timescale and a heat-driven SCO occurs on 10-100 ns timescale. The authors made clear that in this long timescale the current study do support the previous conclusion. The new finding in this paper should then focus on the sub-picosecond SCO on the excited state. In the current manuscript, the description about the two SCOs is rather confusing, and some details about the first SCO are missing. What is the fraction that underwent the ultrafast SCO (I can only find the fraction of the slow SCO)? What happens to these molecules on the intermediate timescale before the second SCO, did they stay in the HS state or flip back to the LS state? Can the authors add a plot to show the SCO fraction as a function of time, covering all the timescales studied (fs to ns)?
2. I think one of the most interesting results from this paper is that they are able to extract the two major reaction coordinates (elongation and rotation of ligands) of the ultrafast SCO, from the fitting of 112 Bragg peaks. In the SI the authors discussed the rational about the choice and presented the workflow of the fitting. However, they did not give any error analysis, neither the uncertainty of the fitting parameters, nor the goodness of the fit. I think it is always important to confirm no underfitting or overfitting is present when fitting to a multidimensional dataset.
3. The SCO only takes 100 meV while the energy in the excited state molecules are 4.6 eV. This is to say that, in the first few ps when the heat is still mostly constraint in the photoexcited molecules, they must be in a very hot HS state. The author did not explicitly discuss this point, and assumed the structure of photoinduced and thermal induced HS has the same structure. I'm concerned about the assumption because of the large temperature difference of the two scenarios. Can they estimate what degree of freedom will be excited in the photoexcited molecules (via IVR for example) and how large the impact will be on the molecular structure?
4. I could seem many oscillations in figure 2d, 3b and 4b. The author completely ignored these features by using a smooth biexponential fitting, I'm wondering whether they are just noisy data or there's actually important physics behind these oscillations.
5. In table 5, the reciprocal space description is only meaningful within the diffraction community. Can they add some real-space description (for example the change of lattice constants a, b, and c) at different time?

Minor points:

- 1 Figure 1a, N is mislabeled as B.
- 2 Title of ref. 34 is incorrect.
3. line 214 radical->radial

Reviewer #3

(Remarks to the Author)

The manuscript presents data collected on a SCO compound thin film sample using femtosecond MeV-UED and XRD. A combined analysis of the two data sets was performed for the first 2 ps to extract the time dependence of the intramolecular Fe-N bond elongation and ligand rotation occurring on ~150 fs/1.5 ps timescales. Within that time frame, no significant unit cell expansion was observed from the XRD data and only a weak DW effect could be observed in the UED data. Within the first ~200 ps, XRD peaks that are sensitive to the Z-direction show changes in peak position and intensity that are assigned to coherent acoustic strain wave dynamics with a ~221 ps oscillation period, that is launched due to the deposition of excess energy within the lattice. Since the strain wave is only observed along the crystallographic Z direction, the expansion along the X, Y directions within tens of picoseconds was predominantly attributed to lattice heating rather than the volume change to accommodate the photoinduced HS fraction (the a-spacing contracts in the HS state). At ~37 ns, the 200 peak shows a contraction of the a-spacing which is used as evidence for a significant thermo-elastic step in the HS fraction on this timescale. The description of the photo-switching dynamics appears to be in line with previous studies on SCO systems and the achievement of a ~63% thermo-elastic step in the HS fraction is consistent with the estimated increase in lattice temperature. While the study provides a comprehensive picture of the photo-switching dynamics, I feel there are some open questions that should be addressed before the study can be considered for publication in Nature Communications.

The manuscript comments on the general benefits of a combined fs UED/XRD analysis at the end of the conclusions section. However, it was not clear to me from the current manuscript if/how that combination was critical to elucidate the structural motions within the first ~2 ps. For instance, I'm wondering how many of the conclusions in the manuscript could be supported from the fs XRD data alone? While the improved time resolution in the UED with respect to previous UED studies on SCO systems is appreciated, the combination of the techniques could also introduce complications for instance with respect to the excitation fraction that was different in both experiments (and therefore photoinduced strain and thermal excess energy). The estimate of the excitation fractions appears to have a relatively high uncertainty as outlined in the Supplementary information and it would be good to understand how the combined analysis is impacted by this. Alternatively, could the experimental limitations mentioned on lines 177-181 (lack of sensitivity along crystallographic Z-

axis) for the UED be alleviated in a different way (e.g. sample prep) instead of performing an XRD experiment at the EuXFEL?

The choice of the structural model strongly builds on several prior studies (including fs UED and XANES studies of SCO systems by some of the authors, refs 13, 14, 26) without presenting alternative structural models. In that regard, it would be good if the authors could discuss more clearly how their new study adds to the bulk of these previous ones regarding the molecular-scale dynamics within the first 2 ps. In that regard, the authors should also add some sort of quantification/visualization of how well the model actually fits the data (e.g. simulated vs. experimental Bragg peak intensities).

According to Supplementary Method 6, the thermo-elastic step magnitude at 37 ns is estimated by neglecting the effect of the lattice temperature and assuming a linear relationship between the HS fraction and the relative change in structure factor with respect to the difference of the HT and LT structure factors. Can the authors comment on the origin of the deviations observed between the 37 ns (red lines) and thermal equilibrium differences (black lines) shown in Fig. 6 and how they potentially impact the analysis? These deviations are somewhat surprising since according to the figure caption, the HT curves were measured at 353 K. This seems to be close to the thermal transition temperature of the compound (which should be mentioned in the manuscript), and the thermal differences should therefore represent a HS fraction not too far from 63%/similar structure compared with the 37 ns lattice rather than a pure HS lattice.

Additional comments

Fig 1a: There appears to be a typo in atoms list.

Figure 5b: The relative XRD intensity changes of peaks sensitive to the crystallographic Z direction show a dip within ~10 ps that appears to be distinct from the oscillation assigned to acoustic strain wave dynamics. Can the authors comment on the origin of this feature?

Figures 2b,c: It appears that there are some subtle deviations between the thermal equilibrium and the 2 ps transient difference diffraction patterns. Are these solely due to the differences between the time-resolved and HT molecular structures visualized in Fig. 4b? Maybe a clarification should be added on/after lines 151-154. Can the same comparison between thermal equilibrium and transient difference diffraction patterns be added in Fig. 3 for the XRD?

Version 1:

Reviewer comments:

Reviewer #1

(Remarks to the Author)

The authors responded well to the reviewers' comments and revised the manuscript accordingly. I would like to support the manuscript to be published.

Reviewer #2

(Remarks to the Author)

The authors presented a very detailed response to my questions and concerns. I have no more issue and would like to recommend the publication of this manuscript on Nature Communications.

Reviewer #3

(Remarks to the Author)

In my opinion, the authors have provided a significantly improved revised manuscript. I feel that the aspects of combined UED/XRD analysis and related novel findings are now presented in a clearer fashion. The additions in the Supplementary Information are helpful. In view of these changes, I'm in favor of publication of this manuscript.

I do have some additional comments for the authors to consider.

Line 141: typos: 'purple atoms', 'pink atoms'

Line 220: "Supplementary Fig. 10" should be "Supplementary Fig.14".

Line 254: I assume "underfitting" should be "overfitting"?

Line 298: I'm confused by the statement that "... the ligand rotation is found to be smaller." Do the horizontal dashed lines in Fig. 4b represent the LS/HS thermal equilibrium values of both Fe-N bond elongation and Fe-ligand rotation coordinates? If yes, it seems to me that both coordinates deviate similarly from the thermal equilibrium values at 2 ps. Maybe the authors can clarify this in the manuscript.

Supplementary Information:

Lines 41/52: Contain different versions of the abbreviation MARIC; I assume the version in line 52 is correct.

Line 71: Should be Supplementary Fig. 3

Line 178: should be Supplementary Fig. 4

Line 198: Fig S16 does not exist, Fig 3c should be Fig 4b.

Lines 335-338/Supplementary Table 1: I'm puzzled by the absence of Bragg peak broadening in the thermal transition. In Response Fig. 3, the thermo-elastic step is depicted on the nanosecond timescale without increasing the HS fraction within the first nanosecond. Therefore, I'm struggling to understand how the <10% photoinduced HS fraction causes a clear increase in the 321 Bragg peak FWHM assigned to structural inhomogeneity (Response Fig. 9) but the thermal transition

(~85% HS fraction at 353 K) does not. Is it possible that the HT data is impacted by additional processes such as e.g. phase separation?

The authors thank the editor and reviewers for careful reading and various comments and questions.
We are glad that the reviewers appreciated the reported comprehensive picture of the photo-
switching dynamics on this spin crossover (SCO) thin film and the complementary nature of the
two distinct diffraction methods based on MeV electrons beams and X-ray free electron lasers
(XFEL). We considered the comments and questions raised by the reviewers carefully and
discussed critical points, including the energy distribution on the picosecond timescale, structural
modelling, and the complementary nature between ultrafast electron diffraction (UED) and X-ray
diffraction (XRD) based on XFELs. We believe all concerns pointed out by the reviewers have
been addressed in the new manuscript. We have prepared point-by-point responses with
corresponding changes in the manuscript as follows. The major changes in the manuscript have
been highlighted as yellow, and details are provided in the individual response.

**Reviewer #1 (Remarks to the Author):**

The manuscript entitled "Capturing ultrafast molecular motions and lattice dynamic response in a
spin crossover thin film using femtosecond electron and X-ray diffraction" by Y. Jiang describes
the complementary ultrafast time-resolved X-ray and electron diffraction experiments on spin
crossover complex crystals. The manuscript is well-written, and the characterization of molecular
crystals with ultrafast X-ray and electron diffraction is quite a hot topic. However, I have several
questions before I support this manuscript to be published in Nature Communications.

The authors appreciate the comments from the reviewer. We hope in our revised manuscript we
have addressed the reviewer's questions and clarified the contribution and applications of our work.

1. On page 3, the authors claim that "However, the exact time evolution and interaction between
molecular structural rearrangement, photothermal heating, change in the unit-cell volume, and
subsequential spin-state switching in (nano-)crystals on ultrafast timescales remains a subject of
ongoing debate." However, what are specific big 'questions' on this material or this class of
materials?

In the context of photoinduced SCO materials, it is essential to understand the molecular-scale
interplay between photoinduced spin-state switching and structural rearrangements. Additionally,
the interaction of these local changes with the crystal lattice and neighboring molecules drives
collective responses, such as bulk lattice deformations and amplification of photoswitched
molecules. However, thermo-elastic effects alone are insufficient, as energy barriers—both local
and bulk—also influence the cooperative electronic and structural transformations. The primary
objective is to elucidate how these couplings impact the efficiency and dynamics of bulk phase
transitions, enabling various photo-switchable functions (optical, conductivity, magnetic,
dielectric) triggered by short light excitation.

In this work, we specifically address this question by monitoring the sub-picosecond molecular
structural dynamics (via Bragg intensities) and the lattice response (via Bragg positions in

reciprocal space). While our results allow only tentative conclusions, they reveal that the ultrafast
dynamics connecting 'what the molecular does' and 'what the lattice sees' are not trivial
juxtaposition of processes originating simultaneously. A single master equation describing all
degrees of freedom out of equilibrium is not possible at this point. The distinct incubation times
for lattice volume expansion and molecular switching highlight this complexity. Although
molecular switching has been discussed in prior studies, our work unambiguously demonstrates it
in a SCO crystal for the first time using two complementary diffraction techniques. Molecular
energy transport in the regime of short distances, short time intervals, and large temperature bursts
remain a theoretical challenge.

Lines 94-103: We have updated the remaining specific question on this material with addition
literature and discussions.

2. The observations (the changes in 150 fs, 2 ps, 40 ps, 37 ns, and acoustic phonon vibration)
correspond well to the previous reports, and the authors completely summarized the dynamics in
this manuscript, which is excellent. Furthermore, the dynamics are observed with state-of-the-art
X-ray free electron laser and MeV time-resolved electron diffraction setup, which is also very
interesting. However, what is newly understood in this manuscript? I believe that the structural
dynamics in 150 fs and 2 ps (so far only in optical methods) seem new, which is still unclear in
the manuscript.

We appreciate the reviewer's recognition of the comprehensive photoinduced dynamics presented
in our study and thank for the feedback on how the new findings are highlighted in this manuscript.
While our results include observations consistent with previous literature to establish the reliability
of our measurements and provide a logical foundation, we also highlight novel findings that
advance the field. In response, we have added a clear summary of the novel observations at 150 fs
and 2 ps, as well as the sequential dynamics on the ps and ns timescales. Detailed clarifications
are provided below:

1. At the 150 fs and 2 ps timescale, we agree with the reviewer that our study, for the first
time with sub-100 fs temporal resolution via diffraction, reveals molecular structural
reorganization at the very short timescale. Specifically, our results demonstrate a two-step
process: an initial stretching of Fe–ligand bond distances, followed by a sequential
rotational movement of the ligand, all occurring within a constant LT unit-cell volume in
under 2 ps.
- 2. Sequential Dynamics on ps and ns timescales:
- a. Compared to previous synchrotron XRD studies, we monitored lattice volume
changes from femtoseconds to nanoseconds with high temporal resolution across
all three crystallographic directions, revealing a strong anisotropy in the dynamics
of the lattice volume change (in-plane vs. out-of-plane).

- b. Talking advantage of the distinct evolution of lattice parameters in crystal structure,
we successfully separated contributions from molecular SCO switching and
photoheating in the lattice dynamics—an unresolved challenge in prior studies of
similar thin films using optical or synchrotron XRD methods, which lacking either
temporal resolution or structural sensitivity.
- c. We demonstrated up to 53% reversible photoexcited molecular fraction in SCO thin
films with a single femtosecond laser pulse at 10 Hz, highlighting its potential for
applications.

We have revised the manuscript to emphasize our new findings on both the ultrafast and later
timescales. The revisions are detailed below:

- 1. Lines 316–318 and lines 323–325: We added a clear description of the novel observations
at the ultrafast timescale.
 - 2. Lines 342–343: We updated the title of the dynamic lattice response section to highlight
our new findings on the interaction between molecular switching and thermal heating
across different timescales.
 - 3. Lines 403–407: We clarified that our new results, based on XRD using XFELs, provide
time-resolved lattice volume expansion along the *X* and *Y* crystallographic directions
associated with strain wave dynamics. These results are enabled by improved temporal
resolution and distinct experimental geometry compared to previous synchrotron scattering
experiments (*Adv. Funct. Mater.* 2403585 (2024), *Commun. Phys.* **5**, 168 (2022)).
 - 4. Lines 408–412: We added a more focused discussion on new findings in the later
timescales, emphasizing the advantages of our experimental approach in temporal
resolution and geometry over previous studies.
 - 5. Lines 413–425: We reorganized the discussion of experimental results in the tens-of-
picoseconds timescale to present our novel observations on the radial position shifts of the
200 and 020 Bragg peaks.
 - 6. Lines 434–437: The discussion of experimental results in the tens-of-nanoseconds
timescale was reorganized for comparison with the tens-of-picoseconds results, revealing
the sequential thermo-elastic SCO and its contribution to lattice volume changes.
 - 7. Lines 459–462: We summarized our observations on the dynamic lattice response and the
respective contributions of molecular switching and lattice heating to lattice volume
changes at different timescales.
 - 8. Lines 472–475: We highlighted the notably high fraction of photo-switched molecules
achieved using a single femtosecond laser pulse.
 - 9. Lines 513–516: The later timescale findings were further emphasized, and we discussed
the impact of these results on the study of photoinduced phase transitions and potential
future applications.

3. The authors used complementary methods with X-ray and electron diffraction measurements.
However, they used the results from X-ray diffraction for most of the discussions. In principle,
both the X-ray and electron diffraction measurements similarly observe the lattice or atomic
dynamics. The results of the electron diffraction measurements are just used to assist the X-ray
diffraction results. Are the results of the electron diffraction measurements needed to support the
discussion in this manuscript? Are the electron diffraction and X-ray diffraction measurements
complementary used in this manuscript?

We realize that the original manuscript may not provide enough details and discussions of the role
of UED data in our analysis. The reviewer is correct that UED and XRD both probe the same
structural processes; however, this combined work also highlights their complementary aspects.
Specifically, under the experimental conditions used, the complementary nature of UED and XRD
lies in differences in q -resolution, q -range, and the number of Bragg peaks from different crystal
planes, which are critical for structural modeling within the first 2 ps. Below, we provide a detailed
discussion of the roles of UED and XRD in supporting our findings and conclusions.

As detailed in the Supplementary Method 4, we used the Pearson correlation coefficient
(Supplementary Equation 4–9), ranging from -1 (indicating a perfect negative linear relationship)
to $+1$ (indicating a perfect positive linear relationship), to quantify the similarity between the
experimentally observed and simulated changes in Bragg peak intensities. In Response Fig. 1, we
show the maximum Pearson correlation coefficient cross-sections for two key structural modes at
$+2$ ps: Fe–ligand elongation (Response Fig. 1a) and ligand rotation (Response Fig. 1b). In
Response Fig. 1a, results from combined UED and XRD Bragg peaks (blue lines) are well behaved,
showing a single, well-defined maximum with high Pearson correlation coefficient, approximately
matching the thermally-induced HS structure (Fig. 4b). In contrast, results using XRD Bragg peaks
alone (red lines) show multiple maxima with comparable and lower correlation values, indicating
reduced accuracy and greater uncertainty in structural refinement. In Response Fig. 1b, the red line
from XRD Bragg peaks alone shows a lower Pearson correlation coefficient and a broader peak
width than the blue line from combined UED and XRD Bragg peaks, reflecting reduced similarity
to the experimental data and larger associated errors. The full-width at half-maximum (FWHM)
of the peak is used to estimate the error bounds in Fig. 3c of the manuscript.

**Response Fig. 1 | Cross-section of the maximum Pearson correlation coefficient of the two**
 **key structural modes at +2 ps. a,** Cross-section of the maximum Pearson correlation coefficient
 of Fe–ligand elongation. **b,** Cross-section of the maximum Pearson correlation coefficient of
 ligand rotation. The blue lines show the Pearson correlation coefficient based on both UED and
 XRD results. The red lines show the Pearson correlation coefficient based on XRD results only.
 **a-b,** the blue marks show the maximums of the Pearson correlation, and the yellow lines show
 prominence and FWHM of the peaks.

Therefore, to refine the molecular model to reveal molecular motions in the first 2 ps, we integrated
 intensity changes from both UED and XRD.

- 1. XRD based on XFELs offers exceptional momentum resolution in reciprocal space ($\sim 10^{-5}$
 \AA^{-1}), but is limited to probing fewer Bragg peaks within the hkl family ($l = 1$ or 2) at lower
 q values ($0.8\text{--}3.0 \text{\AA}^{-1}$), as shown in Fig. 3a. This high momentum resolution is crucial for
 resolving lattice responses to photoexcitation, which UED cannot achieve due to its
 relatively larger uncorrelated beam divergence (*Rev. Mod. Phys.*, **94**, 045004 (2022)), a key
 factor limiting its resolving power. Therefore, XRD forms the basis for most discussions
 on dynamic lattice responses to photoexcitation. However, the smaller unit cells of the
 molecular crystals studied, compared to protein crystals in serial femtosecond
 crystallography (SFX), make structural refinement more challenging due to the fewer
 Bragg peaks within narrower q -range probed as shown in the above discussion.
- 2. UED offers the advantage of efficiently scattering at high-order Bragg peaks, owing to the
 large momentum transfer of MeV electron source. While UED lacks the q -resolution
 needed for lattice dynamics (as shown in Fig. 2b and 2c), it captures a larger number of
 Bragg peaks within the $hk0$ family over an extended q -range ($0.92\text{--}4.7 \text{\AA}^{-1}$) as shown in
 Fig. 2a.
- 3. The extensive q -range and the large number of Bragg peaks from the $hk0$ family in UED
 enhance the statistics for studying coherence structural motions and reduce the risks of

underfitting or overfitting that could arise if relying solely on the fewer Bragg peaks in
XRD, as discussed in Supplementary Method 4. The combined data from UED and XRD
provides a robust foundation for modeling and allows us to confidently assign molecular
structural dynamics.

We have revised the manuscript to emphasize the complementary nature of our UED and XRD
experiments in the structural refinement in the first 2 ps. The revisions are detailed below:

Lines 157–161: We emphasize the advantage of UED in efficiently scattering at high-order Bragg
peaks, enabling extensive q -range coverage and probing a large number of Bragg peaks.

Lines 202–204: We included additional details about the number of Bragg peaks and q -range
coverage in the XRD experiment, along with a comparison to the UED experiment.

Lines 250–261: We expanded the discussion on the UED and XRD data used for molecular
modeling during the first 2 ps.

Lines 524–529: We underscore the complementary roles of the UED and XRD experiments in
molecular modeling on ultrafast timescales in this study.

Supplementary Information Lines 187-206: We added further discussion to compare the fitting
results with UED and XRD and XRD alone.

4. The point is quite interesting: approximately 5% of molecules in the crystal are photoexcited in
a few picoseconds, the energy is distributed to the surrounding molecules, and more than 60% of
molecules in the crystal undergo phase transition. The energy is still at the localized molecular
sites in 2 ps, will be spread out to the lattice in 40 ps, and changes the surrounding molecules to
HS state in 37 ns. Do the molecules have energy in the thermal form or transient intermediate
structural form in 40 ps? All the intensities of X-ray diffraction spots decrease around 100 ps, as
shown in Fig. 5.

We thank the reviewer for highlighting this discussion. Based on prior studies of this compound
(*Adv. Mater.* **31**, 1901361(2019)) and other similar SCO systems (*Nat. Chem.* **7**, 629–633 (2015),
*Phys. Rev. Lett.* **113**, 227402 (2014), *Nat. Commun.* **8**, 15342 (2017), *Nat. Commun.* **11**, 1530
(2020)), the molecules relax to the bottom of the HS state potential energy surface within 2 ps.
Over the next 40 ps, energy dissipates into the lattice, causing lattice expansion for the excited HS
state. This initial constant unit-cell and the sequential lattice expansion in fact cause sequential
molecular rearrangement processes.

- 1. In the first 2 ps, molecules undergo ultrafast rearrangement within a constant unit cell, as
observed here.
- 2. By 40 ps, lattice expansion allows secondary molecular rearrangement, eventually yielding
a fully relaxed HS structure similar to the thermally induced HS state. This secondary
rearrangement, reported in earlier studies (*Chem. - A Eur. J.* **18**, 2051–2055 (2012), *Phys.*
*Chem. Chem. Phys.* **14**, 6192 (2012)), usually occurs in timescale of ns to μ s, beyond the
scope of this work (earliest femtosecond to 100 ps).

The molecules that switch at 40 ns timescale follow a different mechanism compared to photo
switching by laser excitation in sub-2ps. The latter have the relaxation time of 100ns (*Adv. Mater.*
34, 2105468 (2022)), and the relaxation mechanism is well described in (*Chem Phys Chem*, 7,
1127-1135 (2006)). This is longer than the onset of the thermoelastic step, that we probed here at
37 ns, and this step activates LS molecules from the ground state (*J. Mater. Chem. C* 5, 4419
(2017)). The possibility of ground-state molecules to undergo switching depends on the changes
in LS/HS energy barriers caused by volume expansion and crystal temperature increase during the
first step.

The decrease in X-ray diffraction peak intensities in Fig. 5 arises from destructive interferences
caused by periodic atomic displacements in the lattice, associated with breathing acoustic
propagation along the Z crystallographic direction. This interpretation is supported by the fact that
oscillation period matches the film thickness and the reported longitudinal speed of sound.
Refining the structure within this timescale (from -25 to +175 ps) is challenging due to
contributions from structural changes and acoustic phonons in the same timescale. However, at +2
ps, structure refinement is unaffected because lattice acoustic phonons start in later timescale. Our
findings show that these ultrafast molecular rearrangements occur before lattice heating dominates,
enabling a clear characterization of the molecular dynamics during this early stage.

Lines 87–88, we added additional description and references about this molecule rearrangement
process in ns and μ s timescales.

Minor points

1. Fig. 7b is difficult to follow. Please add an arrow for the eye guide or put the figures from left
to right.

In Line 531, we have updated Fig. 7b by adding arrows for the eye guide to present five major
steps of the photoexcitation cycle with explanation in lines 537–546.

2. I would like to see the diffraction intensity changes of electron and X-ray diffraction
measurements in the same index in the supplementary information.

In Response Fig. 2, we present the kinetic traces of the 200 and 020 Bragg peaks from -1 to +4 ps,
captured using both UED and XRD. Due to the significantly different Ewald sphere radii of the
253 keV XFEL and MeV electron sources, only the 200 and 020 peaks in the low- q range are accessible
in both methods. These kinetic traces exhibit similar dynamic features, demonstrating that UED
and XRD capture the same structural dynamics. For the 200 peak, the approximately twofold
difference in relative intensity changes between UED and XRD aligns with the excitation fraction
ratio, confirming linear single-photon excitation conditions as detailed in Supplementary Method
2. For the 020 peak, the apparent bigger differences are attributed to the lower signal-to-noise ratio

in XRD, resulting from Bragg condition poorly matching with keV X-rays compared to MeV
electrons. This observation highlights the wide distribution in excitation fraction estimation based
on Bragg peak intensity changes, underscoring the need to analyze as many Bragg peaks as
possible to improve statistical reliability for investigating coherent molecular motions.

We have added Response Fig. 2 in Supplementary Method 6 with addition discussion in
Supplementary Information Lines 302–310.

**Response Fig. 2 | Kinetic traces of 200 and 020 Bragg peaks from -1 to +4 ps from both UED**
**and XRD measurements.** The solid lines show the results of the global fit to a two-exponential
decay functions (Supplementary Equation 2).

3. In Supplementary Methods 3, there is no Fig. 2e in the main text.

We have updated the correct Figure label in the Supplementary Method 3.

4. In Supplementary Methods 6, which structure factors do the authors use? All the structure
factors they measured?

In Supplementary Fig. 6d in the updated Supplementary information, we present the difference
map of the peaks selected for structural modelling at + 2 ps. We used most of Bragg peaks
measured for the modelling to increase the accuracy of the results and avoid underfitting or
overfitting.

In line 248, we have added this clarification.

**Reviewer #2 (Remarks to the Author):**

In this work, the author studied the ultrafast molecular motion of a spin crossover molecular thin
film using both MeV-UED and XFEL, two state-of-the-art femtosecond diffraction techniques.
They have observed the photoinduced local SCO from LS to HS on sub-picosecond timescale,
followed by a heating process on the 100 ps timescale, then followed by a global SCO on tens of
291 ns timescale. They successfully combined the advantages of two distinct diffraction methods, and
292 built a thorough picture of the structural dynamics across a wide range of temporal scale. I think
the scientific merit for sure meets the standard of Nature Communication. However, I think the
current version of the manuscript still need to be further clarified / improved.

The authors appreciate the positive feedback from the reviewer which helps us further clarify and
improve the manuscript. We have addressed the reviewer's comments carefully. Here are our
responses.

Major points:

1. In the previous synchrotron scattering experiment (ref. 34), it is already established that heating
occurs on 100 ps timescale and a heat-driven SCO occurs on 10-100 ns timescale. The authors
made clear that in this long timescale the current study do support the previous conclusion. The
new finding in this paper should then focus on the sub-picosecond SCO on the excited state.

We thank the reviewer for their thoughtful comments on the sub-picosecond photoinduced SCO
and sequential thermo-elastic driven SCO dynamics discussed in this work. This is the same point
as question 2 from Reviewer #1, and is addressed in our response there.

In the current manuscript, the description about the two SCOs is rather confusing, and some details
about the first SCO are missing. What is the fraction that underwent the ultrafast SCO (I can only
find the fraction of the slow SCO)? What happens to these molecules on the intermediate timescale
before the second SCO, did they stay in the HS state or flip back to the LS state? Can the authors
add a plot to show the SCO fraction as a function of time, covering all the timescales studied (fs
to ns)?

To avoid confusion, we have added a detailed description of these two SCO steps in the solid state
in the introduction. Additionally, we have modified the discussion of dynamics on the nanosecond
timescale, ensuring consistent descriptions for the second thermo-elastic SCO step to clearly
distinguish it from the photoinduced SCO on the ultrafast timescale.

We have added additional descriptions in the results and discussion section as following:
Lines 69–72: we added a clear description of the two SCO steps involved after light excitation on
SCO solids.

Lines 72–73: we updated the description of the first SCO process as the photoinduced SCO
dynamics in ultrafast timescale.

Lines 89–90: we updated the description of the second SCO process as the thermo-elastic driven
SCO in tens of nanoseconds, as a good comparison to the ultrafast photoinduced SCO in sub-
picosecond timescale.

The missing HS fraction for the first photoinduced SCO were actually provided in Supplementary
Method 6 of the previous submission. The estimated SCO fractions are ~6 % for XRD and ~3 %
for UED, estimated by incident laser fluence and relative time-resolved intensity changes in Bragg
peaks. Although different laser fluences were used in UED and XRD due to differences in damage
thresholds (arising from differing repetition rates) and the need to optimize signal-to-noise ratios,
both measurements probe the same structural process under linear single-photon excitation
conditions.

Regarding molecule switching on the intermediate timescale, excited molecules remain in the HS
state, as relaxation to the LS state typically takes 100 ns to 1 μ s. This behavior has been established
by prior optical spectroscopic studies of this compound (*Adv. Mater.* **31**, 1901361 (2019)) and
similar spin-crossover (SCO) systems (*Commun. Phys.* **5**, 168 (2022), *Nat. Mater.* **15**, 606 (2016)).
These studies also confirm that unexcited molecules remain in the LS state until the second thermo-
elastic SCO occurs at approximately 1 ns, owing to the molecular energy barrier between the LS
and HS states.

To illustrate the SCO fraction as a function of time, we present a plot showing the typical evolution
of the HS fraction as a function of time in Response Fig. 3 (also as Supplementary Fig. 9).

- 1. At short timescale (first 2 ps), the fraction of photoswitched HS molecule was determined
by the incident laser fluence (Supplementary Equation 12) and the relative changes in
Bragg peak intensities (Supplementary Equation 13). Both methods yielded consistent
results.
- 2. On intermediate timescales (tens of picoseconds), determining the HS fraction from UED
and XRD results is challenging due to the significant contributions of acoustic phonons on
Bragg peak intensity changes (Fig. 5b). However, optical measurements on the same
sample and similar SCO films indicate that the HS fraction remains constant during this
period. (*Adv. Mater.* **31**, 1901361(2019)), *Nat. Mater.* **15**, 606 (2016), *Commun. Phys.* **5**,
168 (2022).)
- 3. In the longer timescale (tens of nanoseconds), the oscillations in Bragg peak intensity
caused by acoustic phonons are damped so that the HS fraction could be determined from
the structure factors of Bragg peaks, giving a HS fraction of 53 % at +37 ns, due to the
thermo-elastic switching step.

The continuous line in the plot, included as a visual guide, is scaled based on previously measured

optical and synchrotron XRD data (*Adv. Funct. Mater.* 2403585 (2024), *Adv. Mater.* **31**,
1901361(2019)).

**Response Fig. 3 | Typical evolution of the fraction of photoswitched HS molecules.** Two
fractions in ultrafast and tens of nanosecond are determined from laser excitation conditions and
changes in XRD Bragg peak intensities. The solid line, as a visual guide, is scaled based on
previously measured optical and synchrotron XRD work (*Adv. Funct. Mater.* 2403585 (2024), *Adv.*
*Mater.* **31**, 1901361(2019)). The initial jump in HS fraction presents the ultrafast photoinduced
SCO within the first 2 ps, and the second increase presents the sequential thermo-elastic driven
SCO in ns timescale.

Lines 226–227: we added the HS fraction about the first photoinduced SCO.

In Supplementary Information lines 362-374: we discussed the HS fraction at different timescales,
based on estimations from our XRD experimental results and literatures on same or similar SCO
systems, with a new figure showing the fraction as a function of time (Supplementary Fig. 10).

2. I think one of the most interesting results from this paper is that they are able to extract the two
major reaction coordinates (elongation and rotation of ligands) of the ultrafast SCO, from the
fitting of 112 Bragg peaks. In the SI the authors discussed the rationale about the choice and
presented the workflow of the fitting. However, they did not give any error analysis, neither the
uncertainty of the fitting parameters, nor the goodness of the fit. I think it is always important to
confirm no underfitting or overfitting is present when fitting to a multidimensional dataset.

We thank the comment from the reviewer about our few findings during the ultrafast SCO. We
have added more discussions and figures in Supplementary Method 4 to present the error analysis
of our structural refinement.

Similar to the reply to question 3 from reviewer # 2 and Supplementary Method 4, We employed
the Pearson correlation coefficient to evaluate the similarity between the experimentally observed
and simulated changes in Bragg peak intensities. The high correlation coefficient (close to 1) in
Response Fig. 1 demonstrates a strong positive linear relationship. Furthermore, we calculated the
FWHM of the Pearson correlation peak (Response Fig. 4) to estimate the error bar for Fig. 3c in
the manuscript. This analysis provides an error analysis for our structural modeling and clearly
reveals the temporal sequential activation of Fe–ligand elongation followed by ligand rotation on
an ultrafast timescale.

**Response Fig. 4 | Temporal evolution of the Fe–ligand distance (average length of the Fe–N**
**coordination bonds, red open circles) and of the ligand rotation (average N–Fe–N angles,**
**blue open circles) in the ultrafast timescale (from –1 to +2 ps).** Solid lines show the fitting
curves using multi-exponential decay functions (Supplementary Equation 2). Horizontal dashed
lines correspond to the structural parameters in the HT phase (373 K) and LT phase (300 K),
respectively. Error bars represent the full-width at half-maximum (FWHM) of the global
maximum peaks of the Pearson correlation coefficient in reaction coordinate space
(Supplementary Fig. 5).

Furthermore, we added Response Fig. 5 to provide visualization of the similarity between
experimental observed and simulated Bragg peak intensity changes at +2 ps. Response Fig. 5a
shows the static diffraction pattern with colored circles marking the selected Bragg peaks from Fig.
2d. Figures 5b and 5c present the experimental intensity changes from thermally induced and
photoinduced SCO at +2 ps, respectively, while Fig. 5d displays the simulated changes based on

the proposed structures. The simulated changes (Response Fig. 5d) closely align with both the
transient changes at +2 ps (Response Fig. 5c) and the thermally induced changes (Response Fig.
5b), demonstrating strong agreement between the structural model and experimental data. Notably,
while most Bragg peaks show consistent behavior, minor differences are observed, likely due to
the low excitation fraction and limited signal-to-noise ratio for transient diffraction changes in one
selected timepoint (Response Fig. 5c). For instance, the 020 peak (blue circle) shows an intensity
increase in the thermally induced pattern (Response Fig. 5b) but no clear change in the transient
diffraction changes (Response Fig. 5c). However, Fig. 2d reveals a rapid intensity increase
followed by a gradual decline, consistent with the overall behavior.

**Response Fig. 5 | Experimental and simulated changes in the diffraction pattern.** **a**, Static
electron diffraction pattern at 300 K low-temperature (LT). **b**, Difference between the diffraction
patterns of LT and 373 K high-temperature (HT) phases. **c**, Photoinduced changes in the diffraction
pattern measured at +2 ps after photoexcitation at 267 nm at 300 K. **d**, Difference between low
spin (LS) structure determined from X-ray diffraction and excited state structure at +2 ps in the
LT unit cell.

Lines 302 and Lines 313–315: we updated Fig. 4b to show the error bars based on the half-
maximum (FWHM) of the global maximum peaks of the Pearson correlation coefficient in reaction
coordinate space.

Supplementary Information Lines 179–241: we added the above discussion and figures on the
error analysis and comparison of fitting results based on both UED and XRD peaks and XRD
peaks alone.

3. The SCO only takes 100 meV while the energy in the excited state molecules are 4.6 eV. This
is to say that, in the first few ps when the heat is still mostly constraint in the photoexcited
molecules, they must be in a very hot HS state. The author did not explicitly discuss this point, and
assumed the structure of photoinduced and thermal induced HS has the same structure. I'm
concerned about the assumption because of the large temperature difference of the two scenarios.
Can they estimate what degree of freedom will be excited in the photoexcited molecules (via IVR
for example) and how large the impact will be on the molecular structure?

We agree with the reviewer that, as mentioned in the paper, the photon energy at 267 nm is much
greater than the energy difference between the LS and HS states. In the ultrafast timescale (before
2 ps), the deposited energy primarily remains within the photoexcited molecule, and it is only after
2 ps that the energy begins to be transferred to the surrounding lattice, as evidenced by the observed
lattice response in Fig. 3c.

Regarding the comments on molecular structures during the ultrafast timescale and the impact of
photon energy absorption, we have presented parameterized structures of the HS state in the
manuscript, which reflect local molecular switching while maintaining constant unit-cell
parameters. In Fig. 4b, the non-equilibrium HS structures at ultrafast timescales are similar to, but
distinct from, those thermally-induced. Specifically, the Fe–ligand elongation in the photoexcited
state is comparable to the HS molecule in HT, but with smaller ligand rotation. The structure of
the photoinduced HS state at +2 ps thus differs from the thermally-induced SCO mainly due to
internal chemical pressure from neighboring unrelaxed unit cells, which restricts the structural
relaxation of the photoexcited molecules. Additionally, in a related work (*Commun. Phys.* **5**, 168
(2022)), time-resolved mid-IR spectroscopy was used to track ligand motions during vibrational
cooling, highlighting the difference between ultrafast and thermally-induced structural changes.
Supplementary Fig. 4 of that work shows the C=C stretch evolution in the nascent HS state,
indicating a shift to higher energies—characteristic of a state initially carrying excess vibrational
energy and undergoing vibrational cooling (VC). Notably, there is a clear distinction between time-
resolved spectral changes and thermally-induced changes, reflected in the relative differences
before and after the $\Delta\omega$ shift and the displacement of the maximum position of the C=C band.
However, mid-IR spectroscopy lacks the spatial resolution to directly observe these ligand
structural dynamics.

We also appreciate the reviewer's insightful comment about the degrees of freedom excited during
the photoexcitation process. Based on our modelling results, theory calculation of the low-
frequency modes, and prior studies (*Nat. Chem.* **7**, 629–633 (2015), *Phys. Rev. Lett.* **113**, 227402

(2014), *Nat. Commun.* **11**, 1530 (2020), Hauser, A. Light-Induced Spin Crossover and the High-
Spin → Low-Spin Relaxation. in 155–198 (Springer, Berlin, Heidelberg)), the primary modes
excited via intramolecular vibrational relaxation (IVR) include the breathing and bending modes
of the Fe–ligand bonds. These excitations cause the excited molecular structure to expand and
oscillate coherently around the HS equilibrium, losing energy in the process. Our structural
modeling supports this vibrational cooling with ligand rotation being the dominant mode during
this relaxation. These findings and the vibrational cooling times are consistent with SCO systems
pumped with 650 nm (1.9 eV) (*Phys. Rev. Lett.* **113**, 227402 (2014)) and 400 nm (3.1 eV) (*Nat.*
*Chem.* **7**, 629–633 (2015). *Nat. Commun.* **11**, 1530 (2020). *Chem. - A Eur. J.* **22**, 5118–5122 (2016))
light where the photon energy also is much higher than the energy difference between the LS and
HS states. In our case of UV photoexcitation (4.64 eV), while the photoinduced dynamics remain
similar to those observed in other SCO system, the energy that pumped into the system is even
greater per molecule excited. So, a key difference in our study is that energy from UV enables the
switching of a significantly larger fraction of molecules in the nanosecond timescale, from a few
490 percent up to 53% (*Commun. Phys.* **5**, 168 (2022), *Adv. Mater.* **31**, 1901361(2019), *Nat. Mater.*
**15**, 606 (2016)). This highlights the potential for repetitive colossal switching amplitude using a
single femtosecond laser pulse.

Lines 296-301: we added a discussion comparing the structure of the photoinduced HS state at +2
ps with that of the thermally-induced SCO, based on results of our structural modeling.
Supplementary Information lines 138-168, we discussed the degrees of freedom excited during the
photoexcitation process, based on the molecular structure study, theory calculations, and literature.

4. I could seem many oscillations in figure 2d, 3b and 4b. The author completely ignored these
features by using a smooth biexponential fitting, I'm wondering whether they are just noisy data
or there's actually important physics behind these oscillations.

During our data analysis, we also noticed weak oscillations in Fig. 2d of the UED results and Fig.
3b of XRD results. Similar oscillations have been observed in this ultrafast timescale for other
SCO systems using optical spectroscopy (*Phys. Rev. Lett.* **113**, 227402 (2014)) and X-ray
absorption spectroscopy (*Nat. Commun.* **8**, 15342 (2017)), where they were attributed to
vibrational modes of the molecule, particularly the breathing and bending motions of the metal-
ligand bonds. However, after careful analysis and a comprehensive comparison with existing
literature, we conclude that these weak oscillations, given the noise level in our data, cannot offer
sufficient experimental evidence to demonstrate significant physical relevance to the ultrafast
dynamics investigated in this study. Further investigation of the potential oscillations on this or a
similar system would require enhanced instrumental stability, increased statistical data and
complementary ultrafast time-resolved techniques.

1. Oscillations observed in Fig. 2d of the UED results:

- a. The kinetic traces of Bragg peak intensities were fitted using a function that combines
oscillation components with a biexponential decay, as shown in Response Fig. 6. Weak
oscillations with a period of ~ 1 ps might be identified, but they are close to noise level.
To study these oscillations, it requires better statistics from the instrumental stability.
b. Our findings indicate that the photoinduced SCO dynamics in our sample are similar
to those observed in other Fe(II) SCO systems. However, the oscillation period
identified in our study does not match the 300 and 390 fs periods reported for similar
SCO systems (*Phys. Rev. Lett.* **113**, 227402 (2014)), which have been attributed to the
breathing and bending modes of the Fe—N bonds in optical data.
c. As shown in Response Fig. 7, we also conducted Fast-Fourier Transform (FFT)
analysis on the changes in Bragg peak intensities from Fig. 2d after subtracting the
fitted biexponential decay. In Response Fig. 7a, the residual changes in Bragg peak
intensities are displayed after removing the biexponential decay. The residual changes
do not reveal any obvious oscillation, and some fluctuations are observed even before
time zero. In Response Fig. 7b, the FFT results for individual Bragg peaks are shown.
While some frequencies are detected, the results are heavily affected by noise, and no
consistent frequency is observed across most or all peaks.
- 2. Oscillations observed in Fig. 3b of the XRD results: As shown in Response Fig. 8, a similar
analysis was conducted on the changes in Bragg peak intensities from the XRD results.
While some oscillations could be identified after fitting, they are weak and close to noise
level.
3. Possible oscillations in Fig. 4b: Fig. 4b may show oscillations similar to those observed in
Fig. 2d and Fig. 3b, but they are close to the noise level. The bond distances and bond
angles derived from structural modeling in Fig. 4b are based on changes in the intensities
of the UED and XRD results.

Lines 228–233: we added some discussion on the oscillations.

Supplementary Information line 364-414, we added the above as Supplementary Discussion 1.

**Response Fig. 6 | Kinetic traces of UED Bragg peak intensity for selected reflections from -1**
 **to +4 ps.** The solid lines show the results of the fit to a bi-exponential decay functions with
 sinusoidal oscillations.

a)
 **Response Fig. 7 | Fast-Fourier-Transform** a) Changes in Bragg peak intensities after removing
 the fitted exponential decay. b) Amplitude of the Fast-Fourier-Transform of the oscillations from
 different Bragg peaks.

 **Response Fig. 8 | Kinetic traces of XRD Bragg peak intensity for selected reflections from -1 to**
 **+4 ps (left) and 0 to +2ps (right).** The solid lines show the results of the fit to a bi-exponential
 decay functions with sinusoidal oscillations.

5. In table 5, the reciprocal space 1 description is only meaningful within the diffraction community.
Can they add some real-space description (for example the change of lattice constants a, b, and c)
at different time?

We appreciate the suggestion to add some real-space description. The new table directly shows the
changes in lattice constants at selected time delays and helps the discussion on the lattice volume
changes from the molecular switching and lattice heating.

Lines 489: we have converted the radial shift of the Bragg peak positions in q -space into the change
of the lattice constants in Table 1.

Minor points:

1 Figure 1a, N is mislabeled as B.

Corrected.

2 Title of ref. 34 is incorrect.

Corrected. The ref. 34 is ref.15 in the current version, due to changes of the introduction.

3. line 214 radical->radial

Corrected.

**Reviewer #3 (Remarks to the Author):**

The manuscript presents data collected on a SCO compound thin film sample using femtosecond
587 MeV-UED and XRD. A combined analysis of the two data sets was performed for the first 2 ps to
588 extract the time dependence of the intramolecular Fe—N bond elongation and ligand rotation
occurring on ~150 fs/1.5 ps timescales. Within that time frame, no significant unit cell expansion
was observed from the XRD data and only a weak DW effect could be observed in the UED data.
Within the first ~200 ps, XRD peaks that are sensitive to the Z-direction show changes in peak
position and intensity that are assigned to coherent acoustic strain wave dynamics with a ~221 ps
oscillation period, that is launched due to the deposition of excess energy within the lattice. Since
the strain wave is only observed along the crystallographic Z direction, the expansion along the X,
Y directions within tens of picoseconds was predominantly attributed to lattice heating rather than
the volume change to accommodate the photoinduced HS fraction (the a-spacing contracts in the
HS state). At ~37 ns, the 200 peak shows a contraction of the a-spacing which is used as evidence
for a significant thermo-elastic step in the HS fraction on this timescale. The description of the
photo-switching dynamics appears to be in line with previous studies on SCO systems and the
achievement of a ~63% thermo-elastic step in the HS fraction is consistent with the estimated
increase in lattice temperature. While the study provides a comprehensive picture of the photo-
switching dynamics, I feel there are some open questions that should be addressed before the study
can be considered for publication in Nature Communications.

1. The manuscript comments on the general benefits of a combined fs UED/XRD analysis at the
end of the conclusions section. However, it was not clear to me from the current manuscript if/how
that combination was critical to elucidate the structural motions within the first ~2 ps. For instance,
I'm wondering how many of the conclusions in the manuscript could be supported from the fs
XRD data alone?

We agree with the reviewer's comment that we did not sufficiently explain how the combined UED
and XRD techniques help reveal the ultrafast structural dynamics within the first 2 ps. The benefits
of the combined measurements have been addressed to question 3 from Reviewer #1 and question
2 from Reviewer #2.

Regarding the reviewer's comment about the role of XRD, we agree with the reviewer that the
results from XRD are used for most of the discussions, in particular for analyzing the lattice
response on long timescale following photoexcitation. However, for the structural motions within
the first 2 ps, the limited number of Bragg peaks in narrower \$q\$ -range probed by XRD are
challenging for structural refinement. To address this comment, we have provided additional
discussion about the complementary nature of the UED and XRD results in the structural
modelling within the first 2 ps and we have added the following discussions to clarify this point in
the Supplementary Method 4.

2. While the improved time resolution in the UED with respect to previous UED studies on SCO
systems is appreciated, the combination of the techniques could also introduce complications for
instance with respect to the excitation fraction that was different in both experiments (and therefore
photoinduced strain and thermal excess energy).

We acknowledge the use of different excitation fluences in the UED and XRD measurements. The
low repetition rate (10 Hz) in XRD allows for a higher damage threshold and higher pump fluence
compared to the UED measurement (100 Hz) on the same sample. The higher excitation fraction
in XRD results in a better signal-to-noise ratio for radial position shifts and intensity changes in
the Bragg peaks, despite the lower repetition rate. However, we believe the difference in fluence
does not introduce additional complications for the new findings and conclusions of the study,
particularly regarding the structural refinement for the first 2 ps with both UED and XRD data.

1. While it is true that the photoinduced strain and thermal excess energy will be different due
to different pump fluence used in the UED and XRD. However, the UED data under current
experimental conditions lacks the q -resolution to investigate the lattice response. Therefore,
we rely solely on XRD data, which offers much higher q -resolution, for studying the lattice
response between +2 ps and +37 ns.

2. For the ultrafast timescale, both UED and XRD data are used for the structural modelling
during the local molecular rearrangement process. Despite the different pump excitation
fluences in the UED and XRD measurements, both fluences are moderate and remain
within the linear range of single-photon absorption (Supplementary Method 2) (*Nat.*
*Commun.* **11**, 1530 (2020) *Chem. - A Eur. J.* **22**, 5118–5122 (2016). *Angew. Chem. Int. Ed.*
**56**, 7130 (2017)). Thus, we believe that both measurements probe identical dynamics with
a linear response to the excitation fraction. In the structural refinement, simulated changes
in the structure factor are scaled with the excitation fraction (Supplementary Equation 10)
to compute the Pearson correlation (Supplementary Equations 4-9).

3. The estimate of the excitation fractions appears to have a relatively high uncertainty as outlined
in the Supplementary information and it would be good to understand how the combined analysis
is impacted by this.

We acknowledge the uncertainty in the excitation fractions, primarily due to instrumental stability
and the matching of Bragg conditions across different peaks in the two measurements. As noted
in Supplementary Method 6, we also estimated excitation fractions based on the incident laser
fluence, using the optical absorption properties of the studied compound. The results from both
the UED and XRD measurements are consistent. This combined analysis cross-verifies the
excitation fraction estimates, helping to reduce the uncertainty in structural refinement at ultrafast

timescales. The use of multiple Bragg peaks in both UED and XRD data for structural refinement
enhances the spatial resolution of the modeling (Supplementary Fig. 5).

On the other hand, XRD data, which offer better q -resolution, were used to analyze the lattice
dynamics during the thermo-elastic step at longer timescales. Despite a high uncertainty in the
excitation fractions, our results show a notable increase in the fraction of photoexcited HS
molecules at $t = +37$ ns. This uncertainty does not affect the key findings or conclusions of this
study. Additionally, based on the observed lattice expansions at +100 ps (Fig. 5a), known thermal
expansion coefficients, and the heat deposited by the pump laser, we estimate a temperature
increase of approximately 40 K (Supplementary Method 5). This temperature rise correlates well
with a HS fraction of $\sim 50\%$, consistent with the fraction estimated from the Bragg peak intensity
changes.

4. Alternatively, could the experimental limitations mentioned on lines 177-181 (lack of sensitivity
along crystallographic Z -axis) for the UED be alleviated in a different way (e.g. sample prep)
instead of performing an XRD experiment at the EuXFEL?

In our current experimental setup and sample geometry, the limitation of the UED results lies in
its q -resolution and probing only Bragg peaks from the $hk0$ family.

For the sample preparation, as the reviewer mentioned, preparing the sample with the
orthorhombic X or Y crystallographic direction normal to the surface could be a potential solution.
However, in this study, we cannot modify how the crystallites grow preferentially in a direction.
We only successfully deposited the SCO complex by vacuum thermal evaporation to achieve
smooth, dense, highly oriented thin films with the orthorhombic Z direction normal to the surface,
as reported previously (*J. Mater. Chem. C* **5**, 4419, 2017). In particular, this high-quality
preparation allows us to obtain single-crystal diffraction patterns and study Bragg peaks with
similar interplanar spacing separately (Fig. 1c).

We have considered how UED instrument could address the experimental limitations mentioned:

1. To improve the limited q -resolution, reducing the emission aperture size would decrease
the angular spread of electrons, thus enhancing the q resolution. However, this would
reduce beam brightness and significantly increase experimental time, assuming the same
repetition rate. The repetition rate cannot be modified easily since it is constrained by the
time needed for the hot sample to fully recover to the ground state. Longer data collection
698 times would also be impractical due to setup drift (laser stability, time-zero drift, electron
stability).
- 2. While rotation UED measurements could access Bragg peaks from other crystal planes,
they introduce complications such as changes in sample thickness. During rotation, the
sample thickness relative to the probe electrons increases, affecting diffraction quality due

to the short electron mean free path (~200 nm) in MeV electron diffraction. This is less of
a concern for XRD. Furthermore, it is challenging to maintain the same pump laser
incidence angle during rotation process to ensure consistent pump-probe conditions.

In contrast to these limitations on the UED, the XRD experiment in this work provides high q -
resolution and allows access to data from other crystal planes while maintaining the same
experimental geometry and pump-probe conditions. The limitations of the UED are well
compensated by the different characteristics of modern X-ray and electron sources. Notably, this
study integrates two advanced diffraction techniques—UED from a state-of-the-art MeV electron
source and XRD from an XFEL source—into one quantitative algorithm, a novel approach not
previously reported. This work serves as a case study to highlight the strengths, weaknesses, and
complementary nature of these two diffraction methods in investigating ultrafast molecular
dynamics.

5. The choice of the structural model strongly builds on several prior studies (including fs UED
and XANES studies of SCO systems by some of the authors, refs 13, 14, 26) without presenting
alternative structural models. In that regard, it would be good if the authors could discuss more
clearly how their new study adds to the bulk of these previous ones regarding the molecular-scale
dynamics within the first 2 ps. In that regard, the authors should also add some sort of
quantification/visualization of how well the model actually fits the data (e.g. simulated vs.
experimental Bragg peak intensities).

It is true that as the reviewer noted, several prior studies strongly support the choice of the
structural model in this work. However, this choice is also largely based on other key facts,
including the study of the molecular structure at LS and HS states and theoretical calculations on
low-frequency modes. Our choice of the structural mode allowed us to keep the number of degrees
of freedom to a minimum and capture the most important features of the photoinduced dynamics,
while avoiding overfitting and chemically unreasonable structures.

- 1. One major consideration on the choice of structural model is based on the molecular
structure and its changes during the SCO. As shown in Fig. 1a and 1b, the molecule exhibits
high symmetry, justifying a reduction in degrees of freedom. In Figure 4a, we overlaps the
LS and HS state structures, aligned at the iron center, highlighting the structural changes
during the transition can be broken down to symmetric Fe–ligand elongation and ligand
rotation on both HB(tz)₃ ligand.
- 2. Within the classic description of transition-state processes, each molecule would have a
distinct many-body potential energy surface, with distinct modes reflecting the different
degrees of freedom needed to describe the structural dynamics (*Science*, 343, 1108 (2014)).
In practice, molecular structural dynamics are often dominated by a few low-frequency,
large-amplitude modes, which describe the essential dynamics of the system.

- 3. Our theoretical calculations revealed vibrational modes between 70 cm^{-1} and 200 cm^{-1} ,
with results uploaded for reference. Among these, the Fe–ligand symmetric breathing
(Supplementary Movie. 1) and bending modes (Supplementary Movie. 2) correspond to
the Fe–ligand elongation and rotation, contributing most to the structural changes. Thus, at
least two vibrational modes define the reaction coordinate. As noted by the reviewer,
optical and X-ray absorption studies confirm these modes sufficiently describe the
dominant SCO pathway. Other low-frequency modes, such as ligand torsion
(Supplementary Movie. 3) and out of phase Fe–ligand stretching (Supplementary Movie.
4), are reported to be irrelevant during vibrational cooling. (*Nat. Chem.* **7**, 629–633 (2015),
*Phys. Rev. Lett.* **113**, 227402 (2014), *Nat. Commun.* **8**, 15342 (2017), *Nat. Commun.* **11**,
1530 (2020)). Add these modes in the structural modelling could lead to overfitting rather
than presenting important structural dynamics.

In Supplementary Method 4, we have updated the above discussion on the choice of our structural
modes and alternative modes.

To show our novel findings within the first 2 ps in relation to previous studies, it is important to
note that the majority of research on structural dynamics in SCO, and much of our current
understanding, relies heavily on ultrafast XAS and optical spectroscopy. However, resonant
spectroscopic techniques such as Raman, IR, and XAS are unable to resolve ligand rotation as part
of the overall molecular structural dynamics within a crystalline lattice, as the energies associated
with such rotations are too low to be detected. While NMR offers the structural resolution needed
to capture such details, it lacks the temporal resolution necessary to study structural dynamics on
this ultrafast timescale. The direct structure solving with sub-100 fs temporal resolution like these
spectroscopies was not attempted on sub-1 ps timescale.

Compared to the literature, our direct structural results reveal the following:

- 1. The local structural rearrangement of photoexcited molecules occurs unambiguously
within a constant LT unit-cell volume, before unit-cell volume expansion starts. While
molecular switching within the crystal lattice has been studied using various methods, they
(*Adv. Funct. Mater.* 2403585 (2024), *Nat. Commun.* **11**, 1530 (2020), *Chem. - A Eur. J.* **18**,
2051–2055 (2012), *Phys. Chem. Chem. Phys.* **14**, 6192 (2012).) lack either the temporal or
q -resolution necessary to capture local molecular rearrangements in the active crystalline
medium on ultrafast timescales. The high q -resolution of our XRD data confirms the
absence of changes in unit-cell parameters at this short timescale. This critical observation
supports our finding of a non-equilibrium state, where the structural relaxation of
photoexcited molecules is constrained by the chemical pressure exerted by neighboring
unit cells.
- 2. We reveal the sequence of the local structural rearrangement with Fe–ligand elongation
followed by ligand rotation. We exploit the sub-100 fs temporal resolution and high

structural sensitivity of UED and XRD to directly observe the structural dynamics coupled
to the spin transitions, providing the full extent of the nuclear reorganization process.

- a. Due to our sub-100 fs temporal resolution, we resolve the sequential nature of the
Fe–ligand elongation and ligand rotation, which were previously reported as
coupled motions due to the limited (0.4 ± 0.05)-ps instrument response time of keV
UED work.
- b. Our measurements directly probe the atomic motions of the full molecules,
capturing both the Fe–ligand bonds and ligand motions. This contrasts with indirect
spectroscopic studies (*Nat. Chem.* **7**, 629–633 (2015). *Phys. Rev. Lett.* **113**, 227402
(2014), *Coord. Chem. Rev.* **250**, 1642–1652 (2006)) that resolve specific
vibrational modes and X-ray absorption bands (*Phys. Rev. Lett.* **113**, 227402 (2014)
and *Nat. Commun.* **8**, 15342 (2017)) and thus primarily focus on changes in metal-
ligand distances. Our findings highlight ligand rotation as a key mode for stabilizing
the hot photoexcited HS molecule during the IVR process, extending beyond the
traditionally studied reaction coordinate of metal–ligand bond elongation.

For quantification/visualization of the how well the model fits the data, this is the same point as
question 2 from Reviewer #2 and is addressed in our response there.

Lines 316–318 and lines 323–328: we have updated the discussion on the new findings in this
work. In particular, we highlight our findings due to sub-100 fs and high structural sensitivity
compared to previous works which lack either temporal resolution or limited to changes in Fe–N
bond distances.

6. According to Supplementary Method 6, the thermo-elastic step magnitude at 37 ns is estimated
by neglecting the effect of the lattice temperature and assuming a linear relationship between the
HS fraction and the relative change in structure factor with respect to the difference of the HT and
LT structure factors. Could the authors comment on the origin of the deviations observed between
the 37 ns (red lines) and thermal equilibrium (black lines) shown in Fig. 6, and how these
differences may affect the analysis?

We thank the reviewer for raising the concern regarding the deviations observed between the +37
814 ns (red lines) and thermal equilibrium (black lines) in Fig. 6. In Fig.6, we observe the deviations
in radial peak shift and Bragg peak intensity between the +37 ns (red lines) and thermal equilibrium
(black lines).

For the radial peak shift, some peaks align well (020 peak), satisfactorily (200 and 131 peaks),
while the deviation is more significant for (321 peak). This suggests structural inhomogeneities in
the thin film during the spin-transition process, likely due to the X-ray beam probing a range of
SCO crystallites, each with slightly varying transition properties influenced by size, local defects,
or strain. In addition to peak shifts, the transition results in the broadening of Bragg peaks.

Response Fig. 9 and Response Table 1 show the time evolution of the FWHM of the 321 Bragg
 peak and its changes between 313 and 353 K. These inhomogeneities likely cause different degrees
 of broadening and changes in Bragg intensity at thermal equilibrium and at 37 ns, leading to a
 broader distribution when estimating the excitation fraction from different Bragg peaks at +37 ns.
 However, structural analysis within the constant unit cell volume (<2 ps) avoids these additional
 complexities. Regarding the deviation in Bragg peak intensity, we attribute it to the varying HS
 fraction and structural inhomogeneities, as previously mentioned. For the photoinduced case at
 +37 ns, the HS fraction is estimated at 53%, significantly lower than the 85% HS fraction in the
 thermally induced case.

 **Response Fig. 9 | Time evolution of the width of the 321 Bragg peak in FWHM.**
 **Response Table 1 | Changes of peak width of 131 Bragg peak at different time delays.**

Changes of FWHM of 321 Bragg peak [$\times 10^{-4} \text{ \AA}^{-1}$] ($\pm 10^{-4} \text{ \AA}^{-1}$)					
	+2 ps	+85 ps	+175 ps	+37 ns	Thermal SCO
FWHM	0	+3	+3	+5	0

 Lines 447–450: We have added clarification in the manuscript to explain the origin of the
 deviations between the photoinduced difference and thermally-induced difference.
 Supplementary Information 34-349: we explicitly address the potential impact of the big
 deviations on our conclusion.

7. These deviations are somewhat surprising since according to the figure caption, the HT curves
were measured at 353 K. This seems to be close to the thermal transition temperature of the
compound (which should be mentioned in the manuscript), and the thermal differences should
therefore represent a HS fraction not too far from 63%/similar structure compared with the 37 ns
lattice rather than a pure HS lattice.

As pointed out by the reviewer, in the temperature-dependent XRD study, LT and HT curves were
measured at 313 K and 353 K, respectively, which is close to the transition temperature ($T_C = 336$
851 K). We set this small temperature difference to minimize the contributions from the regular thermal
expansion and maximize the contributions from the molecular switching from LS to HS state.
From the previous optical spectroscopic characterization of this sample, the thermal spin transition
is known to be relatively abrupt with 85 % of the transition spanning over this temperature range
[313-353 K] (*Adv. Mater.* **34**, 2105468 (2022)). Thus, the thermal differences in Fig. 6 represent a
change of HS fraction of 85 %, rather than 100 %. Based on this consideration, we corrected the
calculated excitation fraction in the ultrafast and longer ns timescale for XRD results.

On the ultrafast timescale at +2 ps, the excitation fraction is reduced from 8.0% to 6.8%, which
appears to be even more consistent with the value estimated from the laser fluences. In the longer
861 ns timescale, the excitation fraction at +37 ns is corrected from 63 % to 53 %, still indicating a
862 significant increase in the fraction of photoexcited HS molecules during the thermo-elastic
switching process at $t = 37$ ns. On the other hand, as for the UED experiment, the thermal
differences were measured at 300 and 373 K representing a near 100 % HS fraction change.

Lines 451: we updated the estimated fraction of photoswitched HS molecules to 53 % based on
the fact of incomplete molecular switching during the temperature-dependence XRD measurement.
Line 485–488: we mentioned that LT and HT curves are collected at temperatures close to the
transition temperature of the sample, so that these thermal differences represent a HS fraction
change of 85 %.

Supplementary Information lines 297–298: we added description of thermally-induced SCO based
on the previous optical characterization.

Additional comments

Fig 1a: There appears to be a typo in atoms list.

Corrected.

Figure 5b: The relative XRD intensity changes of peaks sensitive to the crystallographic Z
direction show a dip within ~10 ps that appears to be distinct from the oscillation assigned to
acoustic strain wave dynamics. Can the authors comment on the origin of this feature?

For the intensity changes on longer timescales in Fig. 5b, two distinct dynamics are observed in
this work: one on the sub-ps timescale due to photoinduced SCO dynamics and another on the
tens-of-picoseconds timescale due to acoustic phonon dynamics. In fact, the 'dip' feature within 10
ps, as noted by the reviewer, corresponds to a plateau behavior between these two distinct
dynamics. Many Bragg peaks show a significant intensity decrease on the ultrashort timescale,
caused by molecular switching, followed by a plateau (as seen in Fig. 3b). Subsequently,
oscillations emerge due to the propagation of acoustic phonons. This plateau feature is thus visible
in Fig. 5b prior to the onset of oscillations.

Figures 2b,c: It appears that there are some subtle deviations between the thermal equilibrium and
the 2 ps transient difference diffraction patterns. Are these solely due to the differences between
the time-resolved and HT molecular structures visualized in Fig. 4b? Maybe a clarification should
be added on/after lines 151-154. Can the same comparison between thermal equilibrium and
transient difference diffraction patterns be added in Fig. 3 for the XRD?

We recognize subtle deviations between the thermal equilibrium and the +2 ps transient changes
in the diffraction peaks, but their similarity provides structural evidence of photoinduced SCO
from the LS to the HS states.

While most Bragg peaks exhibit similar changes in both cases, some show subtle differences. We
believe these deviations may be from the low excitation fraction and limited signal-to-noise ratio
in the figure to show changes in the diffraction pattern, but the kinetic traces clearly show the
similar behaviors between thermal equilibrium and the +2 ps transient changes. For example, the
020 Bragg peak (blue circle) displays an intensity increase in the thermally induced diffraction
changes (Fig. 2b) but shows no clear change in the +2 ps transient diffraction changes (Fig. 2c).
However, as shown in Fig. 2d, the 020 peak undergoes a rapid intensity increase followed by a
gradual decline.

We attribute the general differences observed in Fig. 2b and Fig. 2c to the difference between the
out-of-equilibrium state at +2 ps and the thermal equilibrium in thermally-induced SCO. These
include differences in molecular structure and lattice volume. At +2 ps, the photoexcited molecules,
with a low excitation fraction, must rearrange within the constraints of the constant LT unit cell.
In contrast, during thermally-induced SCO, most molecules transition to the high-spin (HS) state
within an expanded unit-cell volume.

Regarding the comparison between thermal equilibrium and transient difference diffraction
patterns for the XRD, we present this comparison in Response Fig. 10. Response Fig. 10a displays
the static XRD diffraction pattern, with colored boxes marking the positions of selected Bragg
peaks. Response Fig. 10b shows the difference between the diffraction patterns at 313 K and 353
922 K, while Response Fig. 10c depicts the photoinduced signal measured after a 2-ps time delay. Both

Response Fig. 10b and 10c focus on the selected region of the diffraction pattern highlighted by
the black rectangle in Response Fig. 10a. Additionally, Response Fig. 10d and 10e display the
selected regions of the difference patterns, as indicated by the colored boxes in Response Fig. 10a–
10c.

The thermally induced SCO difference matches the photoinduced SCO difference at +2 ps,
consistent with the UED results shown in Fig. 2b and 2c. However, due to figure number and size
constraints, this comparison is included in the Supplementary Information as Supplementary Fig.
10, with a corresponding discussion in the main manuscript. Furthermore, a more detailed
comparison of selected peaks is presented in Fig. 6 of the main manuscript, which includes
additional data analysis such as averaging over different sample positions, diffraction pattern
calibration, and azimuthal integration. This analysis clearly illustrates the radial peak shifts and
intensity changes of the selected peaks.

Response Fig. 10 | Differences in thermally-induced SCO and photoinduced SCO at +2 ps. a, Static XRD diffraction pattern and color boxes show the positions of the selected Bragg peaks. **b,** difference between the diffraction patterns of the 313 K and 353 K. **c,** photoinduced signal

measured after 2-ps time-delay. **b-c** are the selected area of the diffraction pattern indicated by the
black rectangle in **a**. **d-e** show the selected area of the difference indicated by the color boxes in **a-**
**c**.

In Lines 167–170, we added a clarification to comment the subtle differences between the transient
changes at +2 ps and thermally-induced changes.

In Lines 220–223, we added a discussion of the difference in thermally-induced SCO and
photoinduced SCO at +2 ps.

Supplementary Line 415, we added a comparison between difference in thermally-induced SCO
and photoinduced SCO at +2 ps.

The authors express their gratitude to the editor and reviewers for their thorough review and
valuable feedback. We are pleased to note that the reviewers highly appreciated the revised
manuscript and found our point-by-point responses to their previous questions and concerns
satisfactory. Concerning the additional comments and minor issues raised by Reviewer #3,
we, the co-authors, believe all concerns have been addressed. We also apologize for the typos
and inconsistencies in the manuscript.

Reviewer #1 (Remarks to the Author):

The authors responded well to the reviewers' comments and revised the manuscript
accordingly. I would like to support the manuscript to be published.

Reviewer #2 (Remarks to the Author):

The authors presented a very detailed response to my questions and concerns. I have no more
issue and would like to recommend the publication of this manuscript on Nature
Communications.

Reviewer #3 (Remarks to the Author):

In my opinion, the authors have provided a significantly improved revised manuscript. I feel
that the aspects of combined UED/XRD analysis and related novel findings are now presented
in a clearer fashion. The additions in the Supplementary Information are helpful. In view of
these changes, I'm in favor of publication of this manuscript.

I do have some additional comments for the authors to consider.

Line 141: typos: 'purple atoms', 'pink atoms'

Corrected as 'purple balls' and 'pink balls' in the new line 135.

Line 220: "Supplementary Fig. 10" should be "Supplementary Fig.14".

Corrected.

Line 254: I assume "underfitting" should be "overfitting"?

Corrected.

Line 298: I'm confused by the statement that '... the ligand rotation is found to be smaller.'
Do the horizontal dashed lines in Fig. 4b represent the LS/HS thermal equilibrium values of
both Fe-N bond elongation and Fe-ligand rotation coordinates? If yes, it seems to me that both
coordinates deviate similarly from the thermal equilibrium values at 2 ps. Maybe the authors
can clarify this in the manuscript.

In fact, the horizontal dashed lines in Fig. 4b for Fe–N elongation were incorrectly labelled in
the current version. We thank the reviewers for bringing this to our attention and apologize for
the error.

The issue is from a mistake when changing the y-axis range for the Fe–N bond elongation in
an effort to optimize the figure's presentation of both bond elongation and its error bar during
our second submission. While the values for the temporal evolution of Fe–N bond elongation
are correct, the horizontal dashed line representing the thermal equilibrium value of the Fe–N
bond elongation was not adjusted accordingly and incorrectly labeled. This error led to
inaccurate comparisons between the Fe–N bond elongation at thermal equilibrium and the
photoinduced changes within the first +2 ps. We have addressed this issue by updating Fig. 4b
in the manuscript. Notably, the version of Fig. 4b from our initial submission, which did not
include error bars, was accurate.

We assure the reviewers that these corrections only affect one observation of our results ‘our
finding of a non-equilibrium state, where the structural relaxation of photoexcited molecules
is constrained by the chemical pressure exerted by neighboring unit cells in Fig. 4’ (new lines
291–293 and new lines 314–316), which is consistent with our previous works. They do not
affect the important observation in the ultrafast timescale of this work ‘the local molecular
structural rearrangement occurs within a constant unit-cell volume through a two-step
process, involving initial Fe–ligand bond elongation followed by the ligand rotation.’ (new
lines 319–321).

Supplementary Information:

Lines 41/52: Contain different versions of the abbreviation MARIC; I assume the version in
line 52 is correct.

The abbreviation MARIC in line 41 has been corrected.

Line 71: Should be Supplementary Fig. 3

Corrected.

Line 178: should be Supplementary Fig. 4

Corrected.

Line 198: Fig S16 does not exist, Fig 3c should be Fig 4b.

Corrected.

Lines 335-338/Supplementary Table 1: I'm puzzled by the absence of Bragg peak broadening
in the thermal transition. In Response Fig. 3, the thermo-elastic step is depicted on the
nanosecond timescale without increasing the HS fraction within the first nanosecond.
Therefore, I'm struggling to understand how the <10% photoinduced HS fraction causes a
clear increase in the 321 Bragg peak FWHM assigned to structural inhomogeneity (Response
Fig. 9) but the thermal transition (~85% HS fraction at 353 K) does not. Is it possible that the
HT data is impacted by additional processes such as e.g. phase separation?

No, we believe that the absence of peak broadening in the thermal transition between 313 K
and 353 K is primarily due to the relatively complete SCO switching and the homogeneity of
the sample at both temperatures, as we explained in the previous response and SI. At 313 K
and 353 K, the samples have good homogeneity, with most being in the LS state at 313 K or
the HS state at 353 K. In contrast, a photoinduced HS fraction of less than 10% introduces a
mixture of LS and HS states, leading to reduced sample homogeneity. Thanks to the high
resolution of our X-ray data, we are able to detect these subtle changes.

In our recent publication (*Adv. Funct. Mater.* 2403585 (2024)), we monitored the thermal
evolution of the 002 Bragg peak's width for the same sample (Figure S3(a)). No broadening is
observed after a complete thermally-induced SCO, matching the thermal measurements in this
work (Supplementary Table 1). A notable increase in peak width is only observed near the
phase transition temperature, where approximately half of the sample is in the LS state and
the other half is in the HS state. This mixture of LS and HS states introduces inhomogeneity
in the samples, resulting in significant peak broadening.

In Supplementary Information Lines 331–340, we add further discussions on this peak
broadening and homogeneity nature of the sample to provide a clear explanation.